# Buffer-based Gradient Projection for Continual Federated Learning

**Shenghong Dai**                                             *sdai37@wisc.edu*
*Department of Electrical and Computer Engineering*
*University of Wisconsin-Madison*

**Jy-yong Sohn**                                             *jysohn1108@yonsei.ac.kr*
*Department of Applied Statistics*
*Yonsei University*

**Yicong Chen**                                             *ychen2229@wisc.edu*
*Department of Electrical and Computer Engineering*
*University of Wisconsin-Madison*

**S M Iftekharul Alam**                                   *s.m.iftekharul.alam@intel.com*
*Intel Labs*

**Ravikumar Balakrishnan**                               *ravikumar.balakrishnan@intel.com*
*Intel Labs*

**Suman Banerjee**                                          *suman@cs.wisc.edu*
*Department of Computer Sciences*
*University of Wisconsin-Madison*

**Nageen Himayat**                                          *nageen.himayat@intel.com*
*Intel Labs*

**Kangwook Lee**                                            *kangwook.lee@wisc.edu*
*Department of Electrical and Computer Engineering*
*University of Wisconsin-Madison*

**Reviewed on OpenReview:** *https://openreview.net/forum?id=Xz5IcOizQ6*

## Abstract

Continual Federated Learning (CFL) is essential for enabling real-world applications where multiple decentralized clients adaptively learn from continuous data streams. A significant challenge in CFL is mitigating catastrophic forgetting, where models lose previously acquired knowledge when learning new information. Existing approaches often face difficulties due to the constraints of device storage capacities and the heterogeneous nature of data distributions among clients. While some CFL algorithms have addressed these challenges, they frequently rely on unrealistic assumptions about the availability of task boundaries (i.e., knowing when new tasks begin). To address these limitations, we introduce Fed-A-GEM, a federated adaptation of the A-GEM method (Chaudhry et al., 2019), which employs a buffer-based gradient projection approach. Fed-A-GEM alleviates catastrophic forgetting by leveraging local buffer samples and aggregated buffer gradients, thus preserving knowledge across multiple clients. Our method is combined with existing CFL techniques, enhancing their performance in the CFL context. Our experiments on standard benchmarks show consistent performance improvements across diverse scenarios. For example, in a task-incremental learning scenario using the CIFAR-100 dataset, our method can increase the accuracy by up to 27%. Our code is available at `https://github.com/shenghongdai/Fed-A-GEM`.

# 1 Introduction

Federated Learning (FL) is a machine learning technique that facilitates collaborative model training among a large number of users while keeping data decentralized for privacy and efficient communication. In real-world applications, models trained via FL need the flexibility to continuously adapt to new data streams without forgetting past knowledge. This is critical in a variety of scenarios, such as autonomous vehicles, which must adapt to changes in the surroundings like new buildings or vehicle types without losing proficiency in previously encountered contexts. These real-world considerations make it essential to integrate FL with continual learning (CL) (Shmelkov et al., 2017; Chaudhry et al., 2018; Thrun, 1995; Aljundi et al., 2017; Chen & Liu, 2018; Aljundi et al., 2018), thereby giving rise to the concept of Continual Federated Learning (CFL).

The biggest challenge in CFL, as in CL, is *catastrophic forgetting*, where the model gradually shifts its focus from old data to new data and unintentionally discards previously acquired knowledge. Initial attempts to mitigate catastrophic forgetting in CFL incorporated existing CL solutions at each client of FL, such as storing previous task data in a buffer (a storage area) and reusing them or penalizing the updates of weights that are crucial for preserving the knowledge from earlier tasks. However, recent works (Bakman et al., 2024; Ma et al., 2022; Yoon et al., 2021) have observed that this naïve approach cannot fully mitigate the problem due to two reasons: (i) small-scale devices participating in FL cannot store much of the previous tasks' data due to the limited buffer size, and (ii) while clients update the model to prevent forgetting based on their local data, the non-IID data distributions across clients cause aggregated updates to fail in preventing global catastrophic forgetting, as observed in previous works (Bakman et al., 2024).

To address these challenges, researchers have developed various CFL algorithms. For example, Federated Weighted Inter-client Transfer (FedWeIT) (Yoon et al., 2021) minimizes the interferences by selectively transferring knowledge across clients, while Continual Federated Learning with Distillation (CFeD) (Ma et al., 2022) limits the buffer size and uses knowledge distillation to retain old knowledge. Global-Local Forgetting Compensation (GLFC) (Dong et al., 2022) addresses the local and global forgetting through the gradient compensation and the relation distillation, and Federated Orthogonal Training (FOT) (Bakman et al., 2024) ensures updates for new tasks are orthogonal to old tasks to reduce the interference. Additionally, a data-free approach (Babakniya et al., 2024) uses generative models to synthesize past data distributions.

While these algorithms tackle unique challenges of CFL, they still share a crucial constraint: the need for explicit task boundaries. This means that the learners need to know when tasks change to effectively manage and update the model. Mitigating catastrophic forgetting in practical scenarios where task boundaries are absent throughout the training process, known as *general continual learning* (Buzzega et al., 2020), remains an important open question.

To address these challenges in CFL, we consider leveraging general continual learning methods, specifically A-GEM (Chaudhry et al., 2019), while also considering the existing constraints in FL. To this end, we propose a method called Fed-A-GEM (illustrated in Figure 1), which involves two key components:

1. **Global Buffer Gradients**: Each client computes the local buffer gradient of the global model with respect to its own local buffer data. These local buffer gradients, serving as reference gradients for model updates, are then securely averaged by the server to obtain the aggregated global buffer gradient.

2. **Local Gradient Projection**: Each client updates its local model such that the direction for the model update does not conflict with aggregated global buffer gradient from the previous round of continual learning, ensuring that the information learned by all clients in previous rounds is maintained during every client's model updating process.

**Our contributions**: In this paper, we introduce a simple method for CFL, called Fed-A-GEM. This method utilizes the information learned from previous tasks across clients to effectively mitigate the catastrophic forgetting without having access to task boundaries. Fed-A-GEM can be integrated with existing CFL techniques to enhance their performance. We conduct comprehensive experiments to demonstrate the effectiveness of Fed-A-GEM across various classification tasks in the image and text domains. Fed-A-GEM consistently improves the accuracy and reduces the forgetting across diverse benchmark datasets. We find

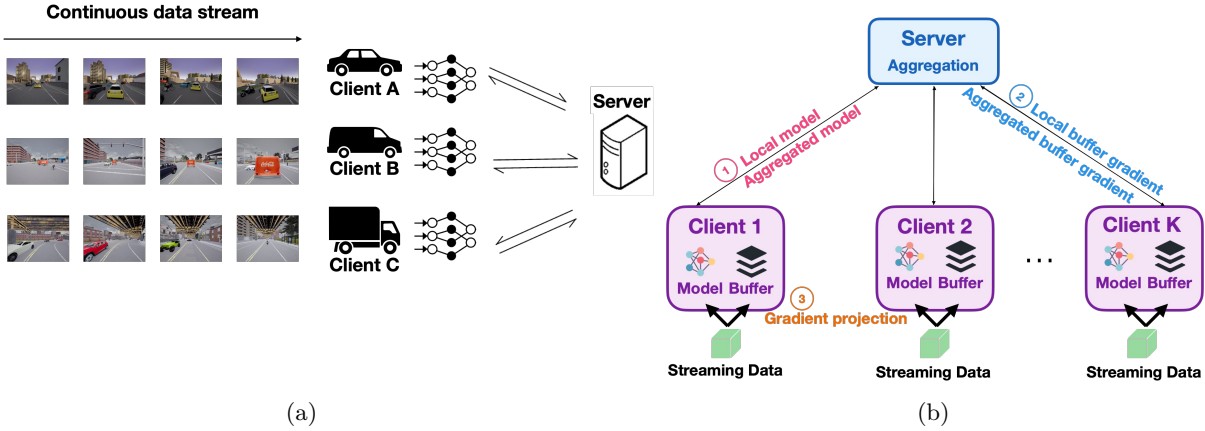

(a)                  (b)

Figure 1: **Overview of Continual Federated Learning.** (a) Challenge of Catastrophic Forgetting: Different driving scenarios encountered by clients in an autonomous vehicle network. Each client faces diverse, dynamic environments, causing vision detection models to forget previous knowledge when learning new tasks. This issue is aggravated by the lack of task boundaries, limited buffer sizes, and non-IID data distribution across clients. (b) System architecture of Fed-A-GEM, showing client-server interactions, local model buffers, and gradient projection mechanisms.

that Fed-A-GEM achieves superior performance without increasing the communication overhead between the server and the clients. Further, we evaluate the robustness of our method considering various buffer sizes, asynchronous environments, and different numbers of tasks and users. Our evaluation also includes an ablation study to examine the role of each key component in the proposed Fed-A-GEM method.

## 2 Related Work

### 2.1 Continual Learning (CL)

CL addresses the problem of learning multiple tasks consecutively using a single model. Catastrophic forgetting (McCloskey & Cohen, 1989; Ratcliff, 1990; French, 1999), where a classifier trained for a current task performs poorly on previous tasks, is a major challenge of CL. Existing approaches in CL can be categorized into regularization-based, architecture-based, and replay-based methods.

**Regularization-based methods** Some CL methods add a regularization term in the loss used for the model update; they penalize the updates on weights that are important for retaining knowledge from previous tasks. Elastic Weight Consolidation (EWC) (Kirkpatrick et al., 2017), Synaptic Intelligence (Zenke et al., 2017), Riemannian Walk (Chaudhry et al., 2018) are methods within this category. EWC uses Fisher information matrix to evaluate the importance of parameters for previous tasks. Besides, Learning without Forgetting (Li & Hoiem, 2017) leverages knowledge distillation to preserve outputs on previous tasks while learning the current task.

**Architecture-based methods** A class of CL methods assigns a subset of model parameters to each task, so that different tasks are learned by different parameters. This class of methods is also known as parameter isolation methods. Some methods including Progressive Neural Networks (Rusu et al., 2016) and Dynamically Expandable Network (Yoon et al., 2018) use dynamic architectures where the architecture changes dynamically as the number of tasks increases. These methods have issues where the number of required parameters grows linearly with the number of tasks. To tackle this issue, fixed network are used in the recent methods including PackNet (Mallya & Lazebnik, 2018), Hard Attention to the Task (Serra et al., 2018) and PathNet (Fernando et al., 2017). These methods maintain a constant architecture regardless of the number of tasks. More recent advancements include Supermasks in Superposition (Wortsman et al., 2020),

which uses a supermask to determine which parameters are used for each task, and DualNet (Pham et al., 2021), which consists of a shared network and a task-specific network.

**Replay-based methods**  To avoid catastrophic forgetting, a class of CL methods employs a replay buffer to save a small portion of the data seen in previous tasks and reuses it in the training of subsequent tasks. One of the early works in this area is Experience Replay (Ratcliff, 1990; Robins, 1995), which uses a memory buffer to store and randomly sample past experiences during training to reinforce previously learned knowledge. In more contemporary studies, Incremental Classifier and Representation Learning (iCaRL) (Rebuffi et al., 2017) stores exemplars of data from previous tasks and adds distillation loss for old exemplars to mitigate the forgetting issue. Deep Generative Replay (Shin et al., 2017) retains the memories of the previous tasks by loading the synthetic data generated by GANs without replaying the actual data for the previous tasks. Gradient based Sample Selection (Aljundi et al., 2019) optimally selects data for replay buffer by maximizing the diversity of samples in terms of the gradient in the parameter space. Gradient Episodic Memory (GEM) (Lopez-Paz & Ranzato, 2017) and its variant Averaged-GEM (A-GEM) (Chaudhry et al., 2019) leverages an episodic memory that stores part of seen samples for each task and use them to compute gradient constraints to prevent forgetting old knowledge. Similarly, Orthogonal Gradient Descent (Farajtabar et al., 2020) stores gradients as opposed to actual data, providing a reference in projection. More recent work include Greedy Sampler and Dumb Learner (Prabhu et al., 2020), Bias Correction (Wu et al., 2019), and Contrastive Continual Learning (Cha et al., 2021). Despite its simplicity, replay-based techniques have shown great performances on multiple benchmarks (Mai et al., 2022; Parisi et al., 2019). Fed-A-GEM leverages this replay-based method, particularly A-GEM, to alleviate forgetting by reusing a portion of previous task data.

**Theoretical efforts on CL**  Recent theoretical advances have deepened our understanding of CL. Bennani et al. (Bennani et al., 2020) establish generalization bounds for both Stochastic Gradient Descent (SGD) and Orthogonal Gradient Descent (OGD) by leveraging the Neural Tangent Kernel (NTK) regime. Their analysis demonstrates that OGD achieves tighter generalization bounds compared to SGD. Kim et al. (Kim et al., 2022) presents a theoretical framework for understanding and improving Class Incremental Learning (CIL). Their work decomposes CIL into two sub-problems: Within-task Prediction and Task-ID Prediction, and shows that Task-ID Prediction is related to Out-of-Distribution detection. Pentina and Lampert (Pentina & Lampert, 2014) bridge the gap between theory and practice in continual learning by proposing a PAC-Bayesian framework. Their work provides theoretical learning bounds on the expected error in future tasks based on the average loss observed in previous tasks. Lee et al. (Lee et al., 2021) provide a rigorous analysis of the relationship between task similarity and two key phenomena in continual learning: forgetting and transfer.

## 2.2  General Continual Learning (GCL)

Existing CL methods often rely on explicit task boundaries to trigger critical actions. For example, some regularization-based methods store neural network responses (e.g., activations) at task boundaries; architecture-based methods update the model architecture after one task is finished; and some replay-based methods update the replay buffer at task boundaries to include samples from the completed task. However, in practical settings, task boundaries are not always clearly defined. This scenario, where sequential tasks are learned continuously without explicit knowledge of task boundaries, is referred to as *general continual learning* (Buzzega et al., 2020; Aljundi et al., 2019; Chaudhry et al., 2019). Dark Experience Replay (DER) (Buzzega et al., 2020) is a notable method in GCL, which retains the network's logits of previous tasks to mitigate forgetting without relying on explicit task boundaries. Gradient Sample Selection (GSS) (Aljundi et al., 2019) deals with the general continual learning and optimizes the diversity of samples in the replay buffer based on gradient feature. A-GEM (Chaudhry et al., 2019) can operate without task boundaries by utilizing a reservoir sampling strategy for its memory buffer. However, most existing works have not thoroughly investigated general continual learning in the context of federated learning. Fed-A-GEM addresses this gap by specifically focusing on the integration and application of general continual learning principles in federated learning.

### 2.3 Federated Learning (FL)

FL enables collaborative training of a model with improved data privacy (McMahan et al., 2017; Kairouz et al., 2021; Lim et al., 2020). FedAvg (McMahan et al., 2017) is a widely used FL algorithm, which averages locally trained models to create a global model. Subsequent works have focused on various challenges in FL, such as improving communication efficiency (Konecnỳ et al., 2016), handling data heterogeneity (Zhao et al., 2018; Karimireddy et al., 2020; Li et al., 2020), and implementing privacy-preserving techniques, including differential privacy (Geyer et al., 2017) and secure aggregation (Bonawitz et al., 2017). Differential privacy aims to ensure that individual data cannot be inferred from the model outputs, while secure aggregation ensures that the individual data remains private when clients share gradients or model updates. FL has been successfully applied in various domains, including autonomous vehicle (Elbir et al., 2022; Dai et al., 2023), healthcare (Chen et al., 2020) and others (Shaheen et al., 2022). However, most existing methods (Li et al., 2020; Shoham et al., 2019; Karimireddy et al., 2020; Li et al., 2019; Mohri et al., 2019) assume static data distribution over time, ignoring temporal dynamics. Our paper addresses this gap by considering temporal dynamics and focusing on a continual federated learning setup.

### 2.4 Continual Federated Learning (CFL)

CFL tackles the problem of learning multiple consecutive tasks in the FL setup. FedProx (Li et al., 2020) and Federated Curvature (FedCurv) (Shoham et al., 2019) aim to preserve previously learned tasks, while FedWeIT (Yoon et al., 2021) and NetTailor (Venkatesha et al., 2022) prevent interference between irrelevant tasks. CFeD (Ma et al., 2022) use surrogate datasets, which are auxiliary datasets approximating past data from previous tasks, to perform knowledge distillation and thus mitigate forgetting. Other methods, including FedCL (Yao & Sun, 2020) and GLFC (Dong et al., 2022), utilize importance weights or class-aware techniques to distill the knowledge from previous tasks. Mimicking Federated Continual Learning (Babakniya et al., 2024) employs generative models to create synthetic data for past distributions. FOT (Bakman et al., 2024) is the state-of-the-art CFL method that aggregates the activation representations of local models and projects the global gradient accordingly on the server side. However, existing CFL methods face several limitations. Some approaches lack scalability as the number of tasks increases (Yoon et al., 2021; Venkatesha et al., 2022). Some methods require surrogate datasets (Ma et al., 2022), which can be difficult and resource-intensive to generate or collect. Furthermore, certain methods incur substantial communication overhead (Yao & Sun, 2020), which can be impractical in federated settings with limited bandwidth. Moreover, many of these methods depend on explicit task boundaries, making them less applicable to general continual learning settings where tasks are not clearly defined (Ma et al., 2022; Dong et al., 2022; Babakniya et al., 2024; Bakman et al., 2024). Our Fed-A-GEM does not rely on explicit task boundaries while maintaining marginal communication overhead.

## 3 Preliminaries

CL focuses on finding a single classifier $f$ (parameterized by $w$) that performs well on $T$ tasks. At time slot $t \in [T]$, the classifier will only have access to the data for task $t$. The notation $[N] := \{1, \ldots, N\}$ is used for a positive integer $N$. The feature-label pair $(x_t, y_t)$ of the samples for task $t$ are drawn from an unknown distribution $D_t$. The goal of CL is to solve the following optimization problem:

$$\min_w \sum_{t=1}^{T} \mathbb{E}_{(x_t, y_t) \sim D_t} \left[ \ell \left( y_t, f \left( x_t; w \right) \right) \right], \tag{1}$$

where $\ell$ is the loss function, and $f(x_t; w)$ is the output of classifier $f$ with parameter $w$, for inputs $x_t$. In practical scenarios, there may be insufficient storage to save all the data seen for the previous tasks. To address this, replay-based methods employ a memory buffer $\mathcal{M}$ that selectively stores a subset of data, which acts as a proxy to summarize past samples and refine the model updates.

Some methods such as DER (Buzzega et al., 2020) use regularization techniques to find the model parameter $w$ that minimizes the loss with respect to the local replay buffer $\mathcal{M}$ as well as current samples. For a given

regularization coefficient $\gamma$, this optimization problem for CL with replay buffers at time $\tau \in [T]$ is:

$$\min_w \; \mathbb{E}_{(x_\tau, y_\tau) \sim D_\tau} \left[ \ell\left(y_\tau, f(x_\tau; w)\right) \right] + \gamma \, \mathbb{E}_{(x_b, y_b) \sim \mathcal{B}} \left[ \ell\left(y_b, f(x_b; w)\right) \right], \tag{2}$$

where $\mathcal{B}$ is a uniform distribution over the samples in buffer $\mathcal{M}$, and $(x_b, y_b)$ are sampled from $\mathcal{B}$. Other methods, such as A-GEM (Chaudhry et al., 2019), introduce a constraint to ensure that the average loss for the data in buffer $\mathcal{M}$ does not increase. Given the model $w_{\tau-1}$ trained on previous tasks, the constrained optimization problem at time $\tau \in [T]$ for A-GEM is represented as:

$$\min_w \mathbb{E}_{(x_\tau, y_\tau) \sim D_\tau} \left[ \ell\left(y_\tau, f\left(x_\tau; w\right)\right) \right],$$
$$\text{s.t.} \quad \mathbb{E}_{(x_b, y_b) \sim \mathcal{B}} \left[ \ell\left(y_b, f\left(x_b; w\right)\right) \right] \le \mathbb{E}_{(x_b, y_b) \sim \mathcal{B}} \left[ \ell\left(y_b, f\left(x_b; w_{\tau-1}\right)\right) \right] \tag{3}$$

An approximate solution to Eq. 3 can be found as follows. Specifically, A-GEM promotes the alignment of the gradient with respect to the current batch of data $(x_\tau, y_\tau)$ and that for the buffer data $(x_b, y_b)$ sampled from the distribution $\mathcal{B}$. Thus, in A-GEM, a proxy of the problem in Eq. 3 is written as:

$$\min_{\tilde{g}} \quad \frac{1}{2} \| g - \tilde{g} \|_2^2,$$
$$\text{s.t.} \quad \tilde{g}^\top g_{\text{ref}} \ge 0 \tag{4}$$

where $g = \nabla_w \, \mathbb{E}_{(x_\tau, y_\tau) \sim D_\tau} \left[ \ell(y_\tau, f(x_\tau; w_{\tau-1})) \right]$ represents the gradient of the loss with respect to the current batch, $g_{\text{ref}} = \nabla_w \, \mathbb{E}_{(x_b, y_b) \sim \mathcal{B}} \left[ \ell(y_b, f(x_b; w_{\tau-1})) \right]$ is the gradient of the loss with respect to the buffer data, and $\tilde{g}$ is the projected gradient we aim to find. The gradient $\tilde{g}$ obtained from solving Eq. 4 is then used to update the model.

For the continual *federated* learning setup where the data is owned by $K$ clients, we use the superscript $k \in [K]$ to denote each client, *i.e.*, client $k$ samples the data from $D_t^k$ at time $t$ and employs a local replay buffer $\mathcal{M}^k$. In the case of using FedAvg (McMahan et al., 2017), each round of the CFL is operated as follows. First, each client $k \in [K]$ performs local updates with $D_t^k$ with the assistance of replay buffer $\mathcal{M}^k$. Second, once the local training is completed, each client sends the model updates to the central server. Finally, the central server aggregates the model updates and transmits them back to clients.

## 4 Fed-A-GEM

We propose Fed-A-GEM, a federated adaptation of the A-GEM method (Chaudhry et al., 2019). Note that Fed-A-GEM is designed to be compatible with various CFL techniques, significantly enhancing their performance in the CFL context. Our approach draws inspiration from A-GEM, which projects the gradient with respect to its own historical data. Building upon this idea, we utilize the global buffer gradient, which is the average buffer gradient across all clients, as a reference to project the local gradient. This allows us to take advantage of the collective experience of multiple clients and mitigate the risk of forgetting the previously learned knowledge in FL scenarios. While reference gradient techniques have been used in Byzantine FL (Cao et al., 2020; Prakash et al., 2020) to defend against malicious updates, we leverage reference gradients to preserve knowledge of old tasks in federated learning. As a replay-based method, Fed-A-GEM maintains a local buffer on each client, which is a memory buffer storing a subset of data sampled from old tasks. The local buffer at client $k$ is denoted by $\mathcal{M}^k$. As the continuous data is loaded to the client, it keeps updating the buffer so that $\mathcal{M}^k$ becomes a good representative of old tasks.

Algorithm 1 provides the overview of our method in the CFL setup, including the process of sharing information (model and buffer gradient) between the server and each client. For each new round $r \in [R]$, where $R$ is the total number of communication rounds per task, the server first aggregates the local models $w^k$ from client $k \in [K]$, getting a global model $w$. Afterwards, the server aggregates the local buffer gradient $g_{\text{ref}}^k$, which is the gradient computed on the global model $w$ with respect to the local buffer $\mathcal{M}^k$, from client $k \in [K]$ to obtain a global buffer gradient $g_{\text{ref}}$. It is worth noting that the term "aggregation" in this context refers to the averaging of locally computed values across all clients. Such aggregation can be securely performed by the central server using secure aggregation, which is denoted as "SecAgg" in Algorithm 1.

---

**Algorithm 1** FedAvg ServerUpdate with Fed-A-GEM

---

Initialize random $w^k$, $\mathcal{M}^k = \{\}$ for all $k$, $g_{\text{ref}} = $ None
**for** each task $t = 1$ **to** $T$ **do**
    **for** each communication $r = 1$ **to** $R$ **do**
        $w^k \leftarrow \texttt{ClientUpdate}(t, w^k, g_{\text{ref}}), \ \ \forall k$
        $w \leftarrow \text{SecAgg}\left(\{w^k\}_{k=1}^K\right)$
        $g_{\text{ref}}^k \leftarrow \texttt{ComputeBufferGrad}(w, \mathcal{M}^k), \ \ \forall k$
        $g_{\text{ref}} \leftarrow \text{SecAgg}\left(\{g_{\text{ref}}^k\}_{k=1}^K\right)$
    **end for**
**end for**
Return the final global model $w$

---

**Algorithm 2** $\texttt{ClientUpdate}(t, w, g_{\text{ref}})$ at client $k$

---

**Input:** Task index $t$, model $w$, global buffer gradient $g_{\text{ref}}$, batch size $\beta$
Load the dataset $\mathcal{D}_t^k$, local buffer $\mathcal{M}^k$
Initialize $n = 0$ at the first task
**for** each batch $\{(x_i, y_i)\}_{i=1}^{\beta}$ in $\mathcal{D}_t^k$ **do**
    $g = \nabla_w \left[\frac{1}{\beta} \sum_{i=1}^{\beta} \ell(y_i, f(x_i; w))\right]$
    $\tilde{g} \leftarrow g - \text{proj}_{g_{\text{ref}}} g \cdot \mathbf{1}(g_{\text{ref}}^\top g \leq 0)$
    $w \leftarrow w - \alpha \tilde{g}$ for some learning rate $\alpha$
    $\texttt{ReservoirSampling}(\mathcal{M}^k, \{(x_i, y_i)\}_{i=1}^{\beta}, n)$
    $n \leftarrow n + \beta$
**end for**
Return $w$

---

**Algorithm 3** $\texttt{ComputeBufferGrad}(w, \mathcal{M}^k)$

---

**Input:** global model $w$, local buffer $\mathcal{M}^k$
$(x_1, y_1) \ldots (x_m, y_m) \leftarrow$ random samples from $\mathcal{M}^k$
$g = \frac{1}{m} \sum_{i=1}^{m} \nabla_w \left[\ell(y_i, f(x_i; w))\right]$
Return $g$

---

Note that here we have two functions used at the client side, `ClientUpdate` and `ComputeBufferGrad`, which are given in Algorithms 2 and 3, respectively. `ClientUpdate` shows how client $k$ updates its local model for task $t$. The client first loads the global model $w$ and the global buffer gradient $g_{\text{ref}}$ which are received from the server in the previous round. It also loads the local buffer $\mathcal{M}^k$ storing a subset of samples for previous tasks, and the data $\mathcal{D}_t^k$ for the current task. For each batch $\{(x_i, y_i)\}_{i=1}^{\beta}$ in $\mathcal{D}_t^k$, the client computes the batch gradient $g$ for the model $w$. The client then compares the direction of $g$ with the direction of the global buffer gradient $g_{\text{ref}}$ received from the server. When the angle between $g$ and $g_{\text{ref}}$ is greater than $90°$, it implies that using the direction of $g$ as a reference for gradient descent may improve performance on the current task, but at the cost of degrading performance on previous tasks. To retain the knowledge on the previous tasks, we do the following: whenever $g$ and $g_{\text{ref}}$ have a negative inner product, we project the gradient $g$ based on the global buffer gradient (which can be considered as a reference) $g_{\text{ref}}$ and remove this component from $g$. The adjusted gradient $\tilde{g}$ is defined as:

$$\tilde{g} = g - \text{proj}_{g_{\text{ref}}} g \cdot \mathbf{1}(g_{\text{ref}}^\top g \leq 0), \qquad (5)$$

where $\text{proj}_{g_{\text{ref}}} g = \frac{g^T g_{\text{ref}}}{g_{\text{ref}}^T g_{\text{ref}}} g_{\text{ref}}$ represents the projection of $g$ onto $g_{\text{ref}}$, and $\mathbf{1}(g_{\text{ref}}^\top g \leq 0)$ is an indicator function that ensures the adjustment is only applied when $g$ and $g_{\text{ref}}$ are in conflicting directions. This projection provides the solution to the constrained optimization problem in Eq. 4, following the idea suggested in A-GEM (Chaudhry et al., 2019). As illustrated in Fig. 2, this projection helps prevent the model updates along the direction that is harming the performance on previous tasks.

After gradient projection, the client updates its local model $w$ by applying the gradient descent step with the updated gradient $\tilde{g}$. Finally, the client updates the contents of the buffer $\mathcal{M}^k$ by using the reservoir sampling (Vitter, 1985) written in Algorithm 4. Reservoir sampling selects a random sample of $|\mathcal{M}^k|$ elements from a local input stream, while ensuring that each element has an equal probability of being included in the sample. One of the advantages of this method is that it does not require any prior knowledge of the data stream. Once the updated local models $\{w^k\}_{k=1}^K$ are securely aggregated on the server using secure aggregation, the server transmits the updated global model $w$ back to each client. Then, each client $k$ computes the local buffer gradient (*i.e.*, the gradient of the model $w$ with respect to the samples in the local buffer $\mathcal{M}^k$) as shown in Algorithm 3.

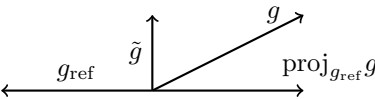

Figure 2: Illustration of the gradient projection in Eq. 5. If the angle between the gradient update $g$ and global buffer gradient (considered as a reference) $g_{\text{ref}}$ is larger than $90°$, we project $g$ to minimize the interference and merely update along the directions of $\tilde{g}$ that is orthogonal to $g_{\text{ref}}$.

After each client computes the local buffer gradient $g_{\text{ref}}^k$, the server allows the use of secure aggregation to combine these local buffer gradients and update the global buffer gradient $g_{\text{ref}}$. This compatibility with secure aggregation enhances the privacy safeguards of our proposed approach, effectively minimizing the risk of data leakage from individual clients. The aforementioned process takes place between each communication and is repeated $R$ times within each task. After traversing $T$ tasks, the final global model $w$ is obtained, as shown in Algorithm 1. Note that the pseudocode describes the FedAvg+Fed-A-GEM process. Fed-A-GEM can be combined with various CFL methods. Specifically, the integration involves using the aggregated buffer gradient computed in Fed-A-GEM as an additional constraint during the local updates in these CFL methods. The detailed examples are given in Appendix E.

## 5 Experiments

In this section, we assess the efficacy of our method, Fed-A-GEM, in combination with various CFL baselines, under non-IID data distribution across clients. To evaluate these methods, we conduct experiments on image classification tasks for benchmark datasets including rotated-MNIST (Lopez-Paz & Ranzato, 2017), permuted-MNIST (Goodfellow et al., 2013), sequential-CIFAR10, and sequential-CIFAR100 (Lopez-Paz & Ranzato, 2017) datasets, as well as a text classification task (Mehta et al., 2023) on sequential-YahooQA dataset (Zhang et al., 2015). We also explore Fed-A-GEM on an object detection task on a streaming CARLA dataset (Dai et al., 2023; Dosovitskiy et al., 2017) in Appendix D. We have further explored an ablation study, which examines each element of our approach, highlighting their roles in performance enhancement. All experiments were conducted on a Linux workstation equipped with 8 NVIDIA GeForce RTX 2080Ti GPUs and averaged across five runs, each using a different seed. For further details and additional results, please refer to Appendix C.

### 5.1 Image Classification

#### 5.1.1 Settings

**Evaluation Datasets.** We evaluate our approach on three CL scenarios: domain incremental learning (domain-IL), class incremental learning (class-IL), and task incremental learning (task-IL). We explain these three types of incremental learning (IL) settings with examples in Appendix A. For domain-IL, the data distribution of each class changes across different tasks. We use the rotated-MNIST (R-MNIST) and permuted-MNIST (P-MNIST) datasets for domain-IL, where each task rotates the training digits by a random angle or applies a random permutation. We create $T = 10$ tasks for domain-IL experiments. For class-IL and task-IL, we use the sequential-CIFAR10 (S-CIFAR10) and sequential-CIFAR100 (S-CIFAR100) datasets, which partition the set of classes into disjoint subsets and treat each subset as a separate task. For instance, in our image classification experiments for class-IL and task-IL, we divide the CIFAR-100 dataset (with $C = 100$ classes) into $T = 10$ subsets, each of which contains the samples for $C/T = 10$ classes. Each task $t \in [T]$ is defined as the classification of images from each subset $t \in [T]$. The difference between class-IL and task-IL is that in the task-IL setup, we assume the task identity $t$ is given at inference time. That is, the model $f$ predicts among the $C/T = 10$ classes corresponding to task $t$. Sequential-CIFAR10 is defined by splitting the CIFAR-10 dataset into $T = 5$ tasks, with each task having two unique classes.

In the FL setup, we assume that the data distribution is non-IID across different clients. Once we define the data for each task, we assign the data to $K$ clients in a non-IID manner. For the rotated-MNIST or permuted-MNIST dataset, each client receives samples for two MNIST digits. To create a sequential-CIFAR10 or sequential-CIFAR100 dataset, we partition the dataset among multiple clients using a Dirichlet distribution (Hsu et al., 2019). Specifically, we draw $\mathbf{q} \sim \text{Dir}(\alpha \mathbf{p})$, where $\mathbf{p}$ represents a prior class distribution over $N$ classes, and $\alpha$ is a concentration parameter that controls the degree of heterogeneity among clients. For our experiments, we use $\alpha = 0.3$, which provides a moderate level of heterogeneity. Communications of models and buffer gradients occur whenever all clients complete the local training for $E$ epochs.

**Architectures and Hyperparameters.** For the rotated-MNIST and permuted-MNIST dataset, we use a simple CNN architecture (McMahan et al., 2017), and split the dataset into $K = 10$ clients. Each client performs local training for $E = 1$ epoch between communications, and we set the number of communication

rounds as $R = 20$ for each task. For the sequential-CIFAR10 and sequential-CIFAR100 datasets, we use a ResNet18 architecture, and divide the dataset into $K = 10$ clients. Each client trains for $E = 5$ epochs between communications, and uses $R = 20$ rounds of communication for each task. During local training, Stochastic Gradient Descent (SGD) is employed with a learning rate of 0.01 for MNIST and 0.1 for CIFAR datasets. Unless otherwise noted, the buffer size is set to $B = 200$, a negligible storage for edge devices.

**Baselines.** We compare the performance of Fed-A-GEM with three types of baselines: 1) *FL*, the foundational FedAvg which trains only on the current task without considering performance on previous tasks; 2) *FL+CL*, which is FedAvg with continual learning solutions applied to clients; and 3) *CFL*, which represents the existing Continual Federated Learning methods.

CL methods we tested include A-GEM (Chaudhry et al., 2019), which aligns model gradients for buffer and incoming data; GPM (Saha et al., 2020), using the network representation/activations approximated by top singular vectors as the old tasks' reference vector; DER (Buzzega et al., 2020), utilizing network output logits for past experience distillation; iCaRL (Rebuffi et al., 2017), which stores a small number of representative examples for each class and counters representation deterioration with a self-distillation loss term; and L2P (Wang et al., 2022), a state-of-the-art approach that instructs pre-trained models to sequentially learn tasks using prompt pool memory spaces.

CFL methods we tested are FedCurv (Shoham et al., 2019), which avoids updating past task-critical weights; FedProx (Li et al., 2020), introducing a proximal weight for global model alignment; CFeD (Ma et al., 2022), using surrogate dataset-based knowledge distillation; and GLFC (Dong et al., 2022), a tripartite method to counteract the forgetting issue: 1) clients adjust the gradients for both old and new classes to ensure balanced updates, 2) clients save the prior model, compute the KL divergence loss between the new and old model outputs (from the last layer), and 3) a proxy server is used to collect perturbed gradient samples from the clients, which are used to select the best previous model. Similar to Fed-A-GEM, FOT (Bakman et al., 2024) projects the gradients on the subspace specified by previous tasks. FedWeIT (Yoon et al., 2021) may not serve as a suitable benchmark given its focus on personalized FL without a global model accuracy to contrast with.

Note that CFeD, GLFC, GPM, iCaRL, and FOT require task boundaries during training. These methods exploit task changes to snapshot the network, where iCaRL further relies on these task boundaries for memory buffer updates. Details of hyperparameters used for the baseline methods are given in Appendix C.7.

**Performance Metrics.** We assess the performance of the global model on the union of the test data for all previous tasks. The average accuracy (measured after training on task $t$) is denoted as $\text{Acc}_t = \frac{1}{t} \sum_{i=1}^{t} a_{t,i}$, where $a_{t,i}$ is accuracy of the global model evaluated on task $i$ after training up to task $t$. Additionally, we measure a performance metric called *forgetting*, which is defined as the difference between the best accuracy obtained throughout the training and the current accuracy (Chaudhry et al., 2018). This metric measures the model's ability to retain knowledge of previous tasks while learning new ones. The average forgetting after seeing $t$ tasks is defined as: $\text{Fgt}_t = \frac{1}{t-1} \sum_{i=1}^{t-1} \max_{j=1,\cdots,t-1} (a_{j,i} - a_{t,i})$. We also compute the Backward transfer (BWT) and Forward transfer (FWT) metrics (Lopez-Paz & Ranzato, 2017), details of which is given in Appendix C.5.

### 5.1.2 Overall Results

Table 1 presents the average accuracy $\text{Acc}_T$ of various methods on image classification benchmark datasets measured upon completion of the final task $T$. For each setting, we compare the performance of an existing method with/without Fed-A-GEM. We observe that the proposed methods (represented by "w/ Fed-A-GEM") improves the base methods ("w/o Fed-A-GEM") in most cases, as seen from the upward arrows indicating performance improvements in Table 1. Similarly, Table 14 in Appendix C.1 compares the forgetting performance $\text{Fgt}_T$, which shows that combining existing methods with Fed-A-GEM reduces the amount of forgetting. Moreover, the analysis in Appendix C.2 demonstrates that Fed-A-GEM consistently outperforms other baselines over time and across tasks. In this Table 1, FL + A-GEM projects the gradient along the local buffer gradient. FL + Fed-A-GEM, on the other hand, projects the gradient along the aggregated buffer gradient. Finally, FL + A-GEM + Fed-A-GEM combines both strategies by first projecting the gradient along the local buffer gradient and then along the aggregated buffer gradient.

Table 1: Average accuracy $Acc_T$ (%) on standard benchmark datasets where "-" indicates experiments we were unable to run, because of compatibility issues (e.g. GLFC and iCaRL in Domain-IL) or the absence of a surrogate dataset (e.g. CFeD on MNIST). The results, averaged over 5 random seeds, demonstrate the benefits of our proposed method in combination with baselines. A buffer of 200 is utilized whenever methods require it. Note that FL+L2P needs an additional pretrained ViT.

| Method | rotated-MNIST (*Domain-IL*) w/o Fed-A-GEM | w/ Fed-A-GEM | sequential-CIFAR10 (*Class-IL*) w/o Fed-A-GEM | w/ Fed-A-GEM | sequential-CIFAR10 (*Task-IL*) w/o Fed-A-GEM | w/ Fed-A-GEM |
|---|---|---|---|---|---|---|
| FL (McMahan et al., 2017) | $68.02^{\pm3.1}$ | $79.46^{\pm4.1}$ (↑**11.44**) | $17.44^{\pm1.3}$ | $18.02^{\pm0.6}$ (↑**0.58**) | $70.58^{\pm4.0}$ | $80.83^{\pm2.0}$ (↑**10.25**) |
| FL+A-GEM (Chaudhry et al., 2019) | $68.34^{\pm5.6}$ | $74.74^{\pm2.3}$ (↑**6.40**) | $17.82^{\pm0.9}$ | $19.44^{\pm0.9}$ (↑**1.62**) | $77.14^{\pm3.1}$ | $83.16^{\pm1.6}$ (↑**6.02**) |
| FL+GPM (Saha et al., 2020) | $74.42^{\pm6.4}$ | $81.12^{\pm2.8}$ (↑**6.70**) | $17.59^{\pm0.4}$ | $20.95^{\pm1.9}$ (↑**3.36**) | $74.50^{\pm3.6}$ | $81.93^{\pm0.3}$ (↑**7.43**) |
| FL+DER (Buzzega et al., 2020) | $57.73^{\pm3.6}$ | $81.33^{\pm3.3}$(↑**23.60**) | $18.44^{\pm3.7}$ | $30.94^{\pm3.8}$ (↑**12.50**) | $69.34^{\pm3.2}$ | $77.99^{\pm0.8}$ (↑**8.65**) |
| FL+iCaRL (Rebuffi et al., 2017) | - | - | $28.54^{\pm3.8}$ | $33.92^{\pm3.0}$ (↑**5.38**) | $80.85^{\pm2.9}$ | $80.09^{\pm4.1}$ (↓**0.76**) |
| FL+L2P (Wang et al., 2022) | $80.90^{\pm3.3}$ | $85.05^{\pm0.7}$ (↑**4.15**) | $28.61^{\pm1.0}$ | $81.86^{\pm7.2}$ (↑**53.25**) | $98.49^{\pm0.1}$ | $98.63^{\pm0.3}$ (↑**0.14**) |
| FedCurv (Shoham et al., 2019) | $68.21^{\pm2.6}$ | $80.53^{\pm4.3}$ (↑**12.32**) | $17.36^{\pm0.7}$ | $17.86^{\pm0.5}$ (↑**0.50**) | $67.77^{\pm1.4}$ | $81.28^{\pm1.1}$ (↑**13.51**) |
| FedProx (Li et al., 2020) | $67.79^{\pm3.2}$ | $78.74^{\pm4.1}$ (↑**10.95**) | $16.67^{\pm2.7}$ | $17.97^{\pm0.8}$ (↑**1.30**) | $69.57^{\pm6.5}$ | $81.23^{\pm1.3}$ (↑**11.66**) |
| CFeD (Ma et al., 2022) | - | - | $16.30^{\pm4.6}$ | $24.07^{\pm8.5}$ (↑**7.77**) | $77.35^{\pm4.6}$ | $79.30^{\pm5.7}$ (↑**1.95**) |
| GLFC (Dong et al., 2022) | - | - | $41.42^{\pm1.3}$ | $41.61^{\pm1.3}$ (↑**0.19**) | $81.84^{\pm2.1}$ | $82.87^{\pm1.0}$ (↑**1.03**) |
| FOT (Bakman et al., 2024) | $70.02^{\pm7.2}$ | $84.14^{\pm4.6}$ (↑**14.12**) | - | - | $73.73^{\pm1.9}$ | $76.94^{\pm2.7}$ (↑**3.21**) |

| Method | permuted-MNIST (*Domain-IL*) w/o Fed-A-GEM | w/ Fed-A-GEM | sequential-CIFAR100 (*Class-IL*) w/o Fed-A-GEM | w/ Fed-A-GEM | sequential-CIFAR100 (*Task-IL*) w/o Fed-A-GEM | w/ Fed-A-GEM |
|---|---|---|---|---|---|---|
| FL | $25.92^{\pm2.1}$ | $34.23^{\pm2.7}$ (↑**8.31**) | $8.76^{\pm0.1}$ | $17.08^{\pm1.8}$ (↑**8.32**) | $47.74^{\pm1.2}$ | $74.71^{\pm0.9}$ (↑**26.97**) |
| FL+A-GEM | $33.43^{\pm1.4}$ | $39.09^{\pm3.5}$ (↑**5.66**) | $8.90^{\pm0.1}$ | $19.53^{\pm1.3}$ (↑**10.63**) | $63.84^{\pm0.8}$ | $74.84^{\pm0.5}$ (↑**11.00**) |
| FL+GPM | $31.92^{\pm3.4}$ | $42.38^{\pm3.5}$ (↑**10.46**) | $8.18^{\pm0.1}$ | $13.32^{\pm1.0}$ (↑**5.14**) | $54.48^{\pm1.4}$ | $65.51^{\pm0.3}$ (↑**11.03** ) |
| FL+DER | $19.79^{\pm1.7}$ | $38.81^{\pm2.0}$ (↑**19.02**) | $13.32^{\pm1.6}$ | $22.96^{\pm3.6}$ (↑**9.64**) | $57.71^{\pm1.2}$ | $65.57^{\pm1.9}$ (↑**7.86**) |
| FL+iCaRL | - | - | $21.76^{\pm1.1}$ | $27.44^{\pm1.2}$ (↑**5.68**) | $69.91^{\pm0.7}$ | $72.83^{\pm0.5}$ (↑**2.92**) |
| FL+L2P | $66.98^{\pm4.6}$ | $69.15^{\pm3.1}$ (↑**2.17**) | $23.12^{\pm1.7}$ | $46.16^{\pm0.4}$ (↑**23.04**) | $94.46^{\pm0.4}$ | $94.91^{\pm0.2}$ (↑**0.45**) |
| FedCurv | $26.00^{\pm2.4}$ | $35.21^{\pm5.1}$ (↑**9.21**) | $8.92^{\pm0.1}$ | $16.67^{\pm0.9}$ (↑**7.76**) | $49.14^{\pm1.6}$ | $74.64^{\pm0.7}$ (↑**25.49**) |
| FedProx | $25.92^{\pm2.5}$ | $35.60^{\pm4.7}$ (↑**9.68**) | $8.75^{\pm0.2}$ | $16.92^{\pm1.4}$ (↑**8.17**) | $47.05^{\pm3.2}$ | $73.95^{\pm0.8}$ (↑**26.89**) |
| CFeD | - | - | $13.76^{\pm1.2}$ | $26.66^{\pm0.3}$ (↑**12.9**) | $51.41^{\pm1.0}$ | $72.20^{\pm0.9}$ (↑**20.79**) |
| GLFC | - | - | $13.18^{\pm0.4}$ | $13.47^{\pm0.7}$ (↑**0.29**) | $49.78^{\pm0.8}$ | $49.20^{\pm1.2}$ (↓**0.58**) |
| FOT | $26.06^{\pm2.0}$ | $29.34^{\pm3.0}$ (↑**3.28**) | - | - | $68.54^{\pm1.5}$ | $74.06^{\pm1.8}$ (↑**5.52**) |

Remarkably, even a simple integration of the basic baseline, FL, with Fed-A-GEM surpassed the performance of most FL+CL and CFL baselines. For instance, in the sequential-CIFAR100 experiment, FL with Fed-A-GEM (17.08% class-IL, 74.71% task-IL) outperformed a majority of the baselines. Specifically, it exceeds the performance of the two advanced CFL baselines: GLFC (13.18% class-IL, 49.78% task-IL) and CFeD (13.76% class-IL, 51.41% task-IL). This underscores the substantial capability of our method in the CFL setting. Importantly, Fed-A-GEM can achieve competitive performance even without utilizing information about task boundaries, unlike CFeD, GLFC, GPM, iCaRL, and FOT.

We also note that the FL+L2P method consistently exhibited the highest accuracy, largely due to the utilization of a pretrained Vision Transformer (ViT) (Dosovitskiy et al., 2020; Zhang et al., 2022), which helps mitigate the catastrophic forgetting. This is why we wrote the numbers in gray with a caveat in the caption. Yet, our approach still managed to achieve significant performance augmentation on top of it. Moreover, we observe that Fed-A-GEM+L2P is most effective in the challenging class-IL cases and least effective in task-IL.

We also compare FL+Fed-A-GEM with the state-of-the-art method, FOT (Bakman et al., 2024). In FOT, at the end of each task, the server aggregates the activations of each local model (computed for local data points) and computes the subspace spanned by the aggregated activations. This subspace is used during the global model update process; the gradient is updated in the direction that is orthogonal to the subspace. While both FOT and Fed-A-GEM project the gradients on the subspace specified by previous tasks, they have two main differences. First, the subspace is defined in a different manner. FOT relies on the representations of local model activations. Fed-A-GEM, on the other hand, relies on the gradient of model computed on its local buffer data. Second, FOT projects the gradient computed at the server side, while Fed-A-GEM projects the gradient computed at each client. We observe that our method, FL+Fed-A-GEM, consistently outperforms FL or FOT across datasets. Additionally, combining Fed-A-GEM with FOT yields superior performance compared to FOT alone, demonstrating the effectiveness of integrating our approach with existing techniques.

We further explored a weighted variant of Fed-A-GEM inspired by FLTrust (Cao et al., 2020), where local updates from clients are weighted according to their similarity with the reference gradient. In this approach, when aggregating client models, we compute a cosine similarity score between each client's local update and the global reference gradient, using these scores as weights during aggregation. Our experiments showed that this weighted aggregation approach did not yield significant improvements over our original method across different datasets and settings. For example, on the sequential-CIFAR10 dataset under the class-IL setting, the weighted variant achieved a marginal improvement of 0.22%. While different weighting schemes and similarity metrics could potentially be explored further, these initial results suggest that our original aggregation approach already captures the essential benefits of gradient-based knowledge preservation.

Table 2: Effect of weighted aggregation on accuracy (%)

| Method | R-MNIST | P-MNIST | S-CIFAR10 | | S-CIFAR100 | |
| | *Domain-IL* | *Domain-IL* | *Class-IL* | *Task-IL* | *Class-IL* | *Task-IL* |
|---|---|---|---|---|---|---|
| Fed-A-GEM | $79.46^{\pm4.1}$ | $34.23^{\pm2.7}$ | $18.02^{\pm0.6}$ | $80.83^{\pm2.0}$ | $17.08^{\pm1.8}$ | $74.71^{\pm0.9}$ |
| Fed-A-GEM w/ weighted Agg | $\mathbf{77.63^{\pm4.8}}$ | $\mathbf{34.28^{\pm4.4}}$ | $\mathbf{18.24^{\pm0.4}}$ | $\mathbf{80.42^{\pm3.8}}$ | $\mathbf{13.46^{\pm0.6}}$ | $\mathbf{74.74^{\pm0.4}}$ |

### 5.1.3 Effect of Buffer Size.

Table 3 reports the performances of baseline CL methods (FL+A-GEM and FL+DER) with/without Fed-A-GEM for different buffer sizes, ranging from 200 to 5120. For most of datasets and IL settings, increasing the buffer size further improves the advantage of applying Fed-A-GEM, by providing more data for replay and mitigating forgetting. However, a finite buffer cannot maintain the entire history of data. In Fig. 3 we reported the effect of buffer size on the accuracy of old tasks. At the end of each task, we measured the accuracy of the trained model with respect to the test data for task 1. We tested on sequential-CIFAR100 dataset, and considered task incremental learning (task-IL) setup. One can observe that when the buffer size $B$ is small, the accuracy drops as the model is trained on new tasks. On the other hand, when $B \geq 100$, the task-IL accuracy for task 1 is maintained throughout the process. Note that training with our default setting $B = 200$ does not hurt the accuracy for task 1 throughout the continual learning process.

Table 3: Impact of the buffer size on $\text{Acc}_T$ (%)

| | | **rotated-MNIST** (*Domain-IL*) | | **sequential-CIFAR100** (*Class-IL*) | | **sequential-CIFAR100** (*Task-IL*) | |
| Buffer Size | Method | w/o Fed-A-GEM | w/ Fed-A-GEM | w/o Fed-A-GEM | w/ Fed-A-GEM | w/o Fed-A-GEM | w/ Fed-A-GEM |
|---|---|---|---|---|---|---|---|
| 200 | | $68.34^{\pm5.6}$ | $74.74^{\pm2.3}$ (↑**6.40**) | $8.90^{\pm0.1}$ | $19.53^{\pm1.3}$ (↑**10.63**) | $63.84^{\pm0.8}$ | $74.84^{\pm0.5}$ (↑**11.00**) |
| 500 | FL+A-GEM | $70.18^{\pm8.7}$ | $78.74^{\pm3.2}$ (↑**8.56**) | $8.87^{\pm0.1}$ | $25.89^{\pm0.9}$ (↑**17.02**) | $64.38^{\pm1.4}$ | $79.35^{\pm0.5}$ (↑**14.97**) |
| 5120 | | $69.97^{\pm3.2}$ | $79.17^{\pm4.3}$ (↑**9.20**) | $8.85^{\pm0.1}$ | $33.30^{\pm2.5}$ (↑**24.45**) | $64.99^{\pm1.5}$ | $84.52^{\pm0.3}$ (↑**19.53**) |
| 200 | | $57.73^{\pm3.6}$ | $87.13^{\pm1.1}$ (↑**29.40**) | $13.32^{\pm1.6}$ | $22.96^{\pm3.6}$ (↑**9.64**) | $57.71^{\pm1.2}$ | $65.57^{\pm1.9}$ (↑**7.86**) |
| 500 | FL+DER | $60.00^{\pm7.2}$ | $88.83^{\pm1.6}$ (↑**28.83**) | $15.44^{\pm1.5}$ | $34.87^{\pm1.7}$ (↑**19.43**) | $60.79^{\pm1.2}$ | $73.53^{\pm1.1}$ (↑**12.74**) |
| 5120 | | $58.63^{\pm3.9}$ | $89.46^{\pm1.2}$ (↑**30.83**) | $18.89^{\pm1.0}$ | $45.76^{\pm3.8}$ (↑**26.87**) | $62.77^{\pm1.5}$ | $83.41^{\pm1.3}$ (↑**20.64**) |
| | | **permuted-MNIST** (*Domain-IL*) | | **sequential-CIFAR10** (*Class-IL*) | | **sequential-CIFAR10** (*Task-IL*) | |
| Buffer Size | Method | w/o Fed-A-GEM | w/ Fed-A-GEM | w/o Fed-A-GEM | w/ Fed-A-GEM | w/o Fed-A-GEM | w/ Fed-A-GEM |
| 200 | | $33.43^{\pm1.4}$ | $39.09^{\pm3.5}$ (↑**5.66**) | $17.82^{\pm0.9}$ | $19.44^{\pm0.9}$ (↑**1.62**) | $77.14^{\pm3.1}$ | $83.16^{\pm1.6}$ (↑**6.02**) |
| 500 | FL+A-GEM | $33.35^{\pm1.0}$ | $42.45^{\pm6.9}$ (↑**9.10**) | $18.39^{\pm0.2}$ | $20.34^{\pm0.6}$ (↑**1.95**) | $78.43^{\pm3.0}$ | $85.95^{\pm0.6}$ (↑**7.52**) |
| 5120 | | $32.72^{\pm1.4}$ | $40.07^{\pm2.5}$ (↑**7.35**) | $16.41^{\pm2.6}$ | $20.64^{\pm2.2}$ (↑**4.23**) | $73.89^{\pm3.3}$ | $86.82^{\pm1.5}$ (↑**12.93**) |
| 200 | | $19.79^{\pm1.7}$ | $43.43^{\pm0.9}$ (↑**23.64**) | $18.44^{\pm3.7}$ | $30.94^{\pm3.8}$ (↑**12.50**) | $69.34^{\pm3.2}$ | $77.99^{\pm0.8}$ (↑**8.65**) |
| 500 | FL+DER | $19.17^{\pm1.6}$ | $43.38^{\pm2.4}$ (↑**24.21**) | $20.81^{\pm3.6}$ | $29.78^{\pm4.3}$ (↑**8.97**) | $71.17^{\pm1.5}$ | $74.98^{\pm3.5}$ (↑**3.81**) |
| 5120 | | $18.57^{\pm1.4}$ | $44.68^{\pm2.4}$ (↑**26.11**) | $34.75^{\pm2.2}$ | $42.38^{\pm4.5}$ (↑**7.63**) | $78.22^{\pm2.3}$ | $81.94^{\pm1.7}$ (↑**3.72**) |

We assume that every client has the same buffer size. If the buffer sizes vary during model training, clients with larger buffers may contribute more diverse data, potentially biasing the model. A possible solution involves using a reweighting algorithm, which we plan to explore in future.

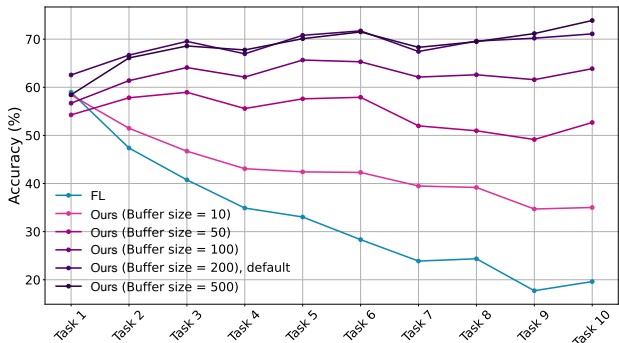

Figure 3: Change in accuracy (%) for task 1 upon completion of subsequent tasks for different buffer sizes, under S-CIFAR100 (Task-IL) setup. Fed-A-GEM with a larger buffer size ($B$) more effectively mitigates forgetting of task 1.

### 5.1.4 Effect of Communication Frequency.

Compared with baseline methods, Fed-A-GEM has extra communication overhead for transmitting the buffer gradients from each client to the server. This means that the required amount of communication is doubled for Fed-A-GEM. We consider a variant of Fed-A-GEM which updates the model and buffer gradient less frequently (i.e., reduces the communication rounds for each task), which has reduced communication than the vanilla Fed-A-GEM. Table 4 reports the performance for different datasets, when the communication overhead is set to 2x, 1x, 0.5x and 0.2x. First, in most cases, Fed-A-GEM with equalized (1x) communication overhead is outperforming FL. In addition, for most of the tested datasets including R-MNIST, P-MNIST and S-CIFAR100, Fed-A-GEM outperforms FL with at most 0.5x communication overhead. This means that Fed-A-GEM enjoys a higher performance with less communication, in the standard CFL benchmark datasets.

Table 4: Effect of communication on accuracy (%), with values in brackets indicating differences from the FL.

| Method | R-MNIST | P-MNIST | S-CIFAR10 | | S-CIFAR100 | |
|---|---|---|---|---|---|---|
| | *Domain-IL* | *Domain-IL* | *Class-IL* | *Task-IL* | *Class-IL* | *Task-IL* |
| FL | $68.02^{\pm3.1}$ | $27.49^{\pm2.0}$ | $17.44^{\pm1.3}$ | $70.58^{\pm4.0}$ | $8.76^{\pm0.1}$ | $47.74^{\pm1.2}$ |
| FL w/ Ours (2× comm) | $\mathbf{79.46^{\pm4.1}}$ (↑11.44) | $\mathbf{35.91^{\pm4.0}}$ (↑8.42) | $\mathbf{18.02^{\pm0.6}}$ (↑0.58) | $\mathbf{80.83^{\pm2.0}}$ (↑10.25) | $\mathbf{17.08^{\pm1.8}}$ (↑8.32) | $\mathbf{74.71^{\pm0.9}}$ (↑26.97) |
| FL w/ Ours (equalized comm) | $75.63^{\pm3.9}$ (↑7.61) | $34.96^{\pm3.2}$ (↑7.47) | $16.65^{\pm1.0}$ (↓0.79) | $78.79^{\pm2.8}$ (↑8.21) | $13.62^{\pm0.6}$ (↑4.86) | $73.96^{\pm0.4}$ (↑26.22) |
| FL w/ Ours (0.5× comm) | $76.05^{\pm4.0}$ (↑8.03) | $29.75^{\pm4.6}$ (↑2.26) | $14.30^{\pm1.3}$ (↓3.14) | $66.90^{\pm3.6}$ (↓3.68) | $13.09^{\pm0.5}$ (↑4.33) | $69.96^{\pm0.6}$ (↑22.22) |
| FL w/ Ours (0.2× comm) | $70.59^{\pm4.7}$ (↑2.57) | $15.51^{\pm2.7}$ (↓11.98) | $13.37^{\pm2.6}$ (↓4.07) | $59.75^{\pm6.4}$ (↓10.83) | $13.59^{\pm0.9}$ (↑4.83) | $59.31^{\pm1.6}$ (↑11.57) |

### 5.1.5 Effect of Computation Overhead.

Computation overhead is also an important aspect to consider and we have conducted experiments on the actual wall-clock time measurements. Taking a CIFAR100 experiment as an example, the running time for 200 epochs for FedAvg on our device is 4068.97s. When Fed-A-GEM, which is built on top of FedAvg, was used, it ran for an additional 293.26s. This indicates that it ran 7.2% longer over the same 200 epochs. Thus, adding Fed-A-GEM has negligible increment in the required computational overhead. The secure aggregation implementation follows the approach described in this work (Bonawitz et al., 2017). Our results indicate that secure aggregation introduces an additional overhead of 172.01 seconds, representing a 4.23% increase in training time. This overhead includes the generation of cryptographic key pairs for all participating clients, and the application of masks to client weights during both model secure aggregation and buffer gradient secure aggregation.

The time consumed by Fed-A-GEM can be divided into two parts: (i) computing the global reference gradient after each FedAvg, and (ii) projecting the gradient. For part (i), the computational complexity of computing the global reference gradient for each client involves sampling from the buffer, computing the gradient for each sample, and averaging these gradients. This process has a complexity of $O(mP)$, where $m$ is the

number of samples in the buffer and $P$ is the total number of parameters. In our experiment, the reference gradient computation was performed 200 times, taking a total of 49.07s. For part (ii), the gradient projection was performed on 109,471 batches, which is 68.38% of the total batches, taking a total of 244.19s. The computational complexity of the gradient projection step, which involves operations such as dot products, scalar multiplication, vector scaling, and subtraction, is $O(P)$, where $P$ is the total number of parameters.

### 5.1.6 Asynchronous Task Boundaries.

In our previous experiments, we assumed synchronous task boundaries where clients finish tasks at the same time. However, in many real-world scenarios, different clients finish each task asynchronously. Motivated by this practical setting, we conducted experiments in an asynchronous task boundary setting on sequential-CIFAR100. The CIFAR-100 dataset was divided into 10 tasks, each containing 10 unique classes, totaling 100 classes. Data was distributed across 10 clients in a non-IID manner using a Dirichlet distribution with $\alpha = 0.3$. Figure 4 illustrates the task distribution across users. In the asynchronous setup, each client processes exactly 500 samples at a time, regardless of task boundaries. The vertical dashed lines indicate 500-sample intervals. As shown by the colored bars in Figure 4, clients have varying amounts of data per task, thus clients may work on different classes and be at different stages of a task compared to others at any given time. This setup more closely aligns with our general continual learning settings, when the task boundary is unknown. Table 5 shows the accuracy of each method averaged over $T$ tasks after finishing all training, under the asynchronous setting. Similar to the synchronous case, Fed-A-GEM improves the accuracy of baseline methods including FL+A-GEM and FL+DER. Notably, we have a better performance in the asynchronous setting (see Table 5) compared with the synchronous setting (see Table 1). For example, under the sequential-CIFAR100 task, the FL+DER+Fed-A-GEM method achieves 72.02% accuracy for the asynchronous case while achieving 65.57% for the synchronous case. This might be because, in the asynchronous setting, some clients receive new tasks earlier than others, which allows the model to be exposed to more diverse data for each round, thus reducing the forgetting.

Table 5: Acc$_T$ (%) for asynchronous task boundaries on the sequential-CIFAR100 dataset.

| Method | Class-IL | Task-IL |
|---|---|---|
| FL | $16.22^{\pm 1.2}$ | $59.04^{\pm 1.7}$ |
| FL+A-GEM | $16.92^{\pm 1.0}$ | $69.41^{\pm 1.3}$ |
| FL+A-GEM+Ours | $30.74^{\pm 1.5}$ | $\mathbf{77.70^{\pm 0.4}}$ |
| FL+DER | $31.95^{\pm 2.6}$ | $68.28^{\pm 1.5}$ |
| FL+DER+Ours | $\mathbf{36.29^{\pm 1.0}}$ | $72.02^{\pm 0.7}$ |

### 5.1.7 Effect of the Number of Tasks.

As shown in Table 6, we have conducted experiments with different numbers of tasks for each dataset. For CIFAR100, we experimented with task numbers 5 and 10, while for CIFAR10 we tested with task numbers 2 and 5. Our results in Table 6 consistently demonstrate that the Fed-A-GEM algorithm provides a significant improvement in performance across all these different task numbers. An interesting observation is that as the number of tasks increases, Fed-A-GEM have better performance improvement to baseline. For example, under the sequential-CIFAR100 task with 10 tasks, FL+Fed-A-GEM achieves a performance improvement of 26.97% compared to the baseline, while with 5 tasks, it achieves an improvement of 18.81%. This is because a higher number of tasks increases the likelihood of data distribution shifts and therefore the problem of catastrophic forgetting becomes more prominent. As such, Fed-A-GEM, designed to handle this issue, has more opportunities to improve the learning process in such scenarios. This might also partly explain why, in Table 1, Fed-A-GEM shows a generally higher improvement over the baselines on the sequential-CIFAR100 dataset compared to the sequential-CIFAR10.

### 5.1.8 Effect of the Number of Users.

We also conducted experiments to assess the scalability of Fed-A-GEM by increasing the client count to $K = 20$. Table 7 shows the results for $K = 20$ clients. These results demonstrate that Fed-A-GEM consistently

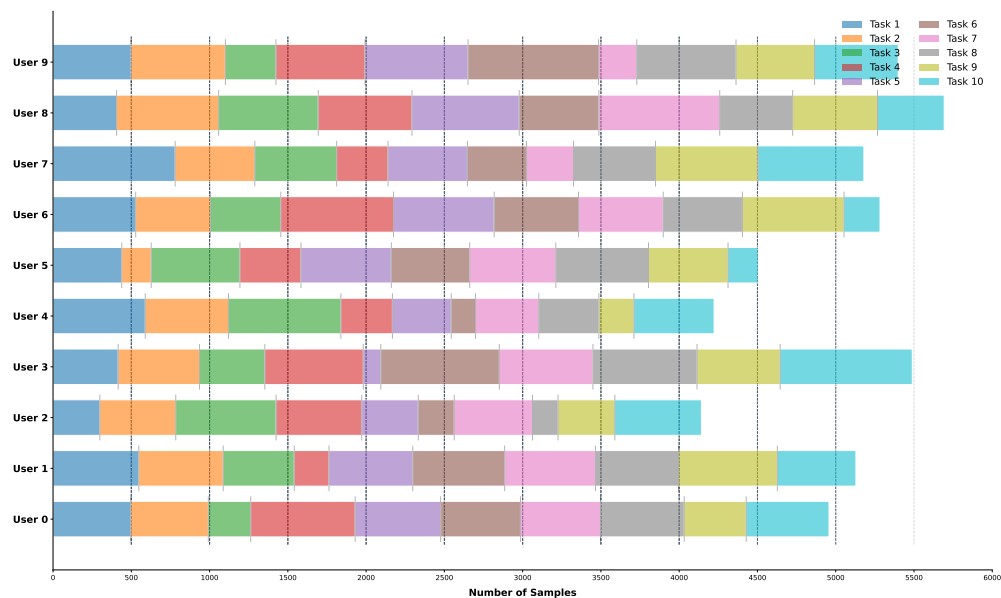

Figure 4: Task distribution across users for the asynchronous setting with CIFAR-100. The dataset is divided into 10 tasks (different colors). Data is distributed non-IID across 10 clients. Clients process 500 samples at a time **(vertical dashed lines)**, progressing through tasks asynchronously.

Table 6: Average accuracy $\text{Acc}_T$ (%) across various task numbers.

| (# of Task, # of Classes per Task) | sequential-**CIFAR10** (*Class-IL*) | | sequential-**CIFAR10** (*Task-IL*) | |
|---|---|---|---|---|
| | FL | FL w/ Fed-A-GEM | FL | FL w/ Fed-A-GEM |
| (2, 5) | $43.53^{\pm0.8}$ | $44.05^{\pm0.8}$ (↑**0.52**) | $75.54^{\pm0.6}$ | $77.52^{\pm0.8}$ (↑**1.98**) |
| (5, 2) | $17.44^{\pm1.3}$ | $18.02^{\pm0.6}$ (↑**0.6**) | $70.58^{\pm4.0}$ | $80.83^{\pm2.0}$ (↑**10.25**) |
| | sequential-**CIFAR100** (*Class-IL*) | | sequential-**CIFAR100** (*Task-IL*) | |
| | FL | FL w/ Fed-A-GEM | FL | FL w/ Fed-A-GEM |
| (5, 20) | $16.49^{\pm0.3}$ | $22.71^{\pm0.9}$ (↑**6.22**) | $50.60^{\pm0.9}$ | $69.41^{\pm0.8}$ (↑**18.81**) |
| (10, 10) | $8.76^{\pm0.1}$ | $17.08^{\pm1.8}$ (↑**8.32**) | $47.74^{\pm1.2}$ | $74.71^{\pm0.9}$ (↑**26.97**) |

improves the performance of baselines, across different numbers of clients. When the number of users increases from $K = 10$ to $K = 20$, methods without Fed-A-GEM generally perform worse because each user has less data. However, Fed-A-GEM performs better with more users in several cases because it uses the increased diversity of data across users to guide learning and preserve knowledge more effectively. This shows Fed-A-GEM's ability to turn the challenge of more users into an advantage. Additionally, we evaluated a real-world scenario where only a random subset of clients participates in training during each round. Moreover, inspired by the client incremental setup described in the GLFC paper, we simulated a dynamic environment where new clients are gradually introduced. Detailed information and results are available in Appendix C.3.

### 5.1.9 Effect of Larger Datasets.

We have conducted experiments on the Tiny-ImageNet dataset, which offers a more naturally diverse and challenging setting for continual learning. For the Tiny-ImageNet dataset, we divided it into 10 tasks, with each task containing 20 unique classes. Given the increased complexity and scale of the dataset, we increased the number of local epochs to 20. We used a pre-trained ResNet-18 model for our experiments. The results are shown in the Table 8. Our method demonstrates significant improvements in both Class-IL and Task-IL accuracy compared to the baselines on this more large-scale dataset. For example, in the Task-IL setting, FL+A-GEM+Fed-A-GEM achieves 50.27% accuracy, while the baseline FL achieves 32.41%.

Table 7: The $\text{Acc}_T$ (%) performance measured when we have $K = 20$ users.

| Method | rotated-MNIST (*Domain-IL*) | | sequential-CIFAR10 (*Class-IL*) | | sequential-CIFAR10 (*Task-IL*) | |
|---|---|---|---|---|---|---|
| | w/o Fed-A-GEM | w/ Fed-A-GEM | w/o Fed-A-GEM | w/ Fed-A-GEM | w/o Fed-A-GEM | w/ Fed-A-GEM |
| FL | $62.45^{\pm 8.5}$ | $76.01^{\pm 4.6}$ (↑**13.56**) | $16.44^{\pm 1.4}$ | $15.82^{\pm 1.7}$ (↓**0.62**) | $68.18^{\pm 5.3}$ | $73.45^{\pm 4.3}$ (↑**5.27**) |
| FedCurv | $62.57^{\pm 8.3}$ | $76.46^{\pm 4.1}$ (↑**13.89**) | $17.31^{\pm 0.6}$ | $14.64^{\pm 3.1}$ (↓**2.67**) | $67.33^{\pm 3.3}$ | $70.31^{\pm 3.7}$ (↑**2.98**) |
| FedProx | $62.14^{\pm 8.6}$ | $75.84^{\pm 4.4}$ (↑**13.70**) | $16.37^{\pm 1.1}$ | $16.15^{\pm 1.3}$ (↓**0.22**) | $66.24^{\pm 1.4}$ | $74.79^{\pm 3.9}$ (↑**8.55**) |
| FL+A-GEM | $67.66^{\pm 8.0}$ | $78.10^{\pm 3.6}$ (↑**10.44**) | $16.15^{\pm 1.9}$ | $17.36^{\pm 0.8}$ (↑**1.21**) | $72.39^{\pm 3.4}$ | $80.61^{\pm 2.6}$ (↑**8.22**) |
| FL+DER | $57.33^{\pm 3.2}$ | $87.84^{\pm 1.5}$ (↑**30.51**) | $17.13^{\pm 2.3}$ | $19.18^{\pm 3.7}$ (↑**2.05**) | $70.82^{\pm 1.9}$ | $77.04^{\pm 2.5}$ (↑**6.22**) |
| Method | permuted-MNIST (*Domain-IL*) | | sequential-CIFAR100 (*Class-IL*) | | sequential-CIFAR100 (*Task-IL*) | |
| FL | $20.26^{\pm 1.6}$ | $20.67^{\pm 4.7}$ (↑**0.41**) | $8.61^{\pm 0.1}$ | $17.47^{\pm 1.1}$ (↑**8.86**) | $50.00^{\pm 1.6}$ | $76.29^{\pm 0.8}$ (↑**26.29**) |
| FedCurv | $20.25^{\pm 1.9}$ | $23.30^{\pm 5.7}$ (↑**3.05**) | $8.93^{\pm 0.0}$ | $19.42^{\pm 1.1}$ (↑**10.49**) | $49.83^{\pm 1.4}$ | $79.58^{\pm 0.6}$ (↑**29.75**) |
| FedProx | $20.19^{\pm 1.4}$ | $23.78^{\pm 5.2}$ (↑**3.59**) | $8.88^{\pm 0.1}$ | $18.86^{\pm 1.0}$ (↑**9.98**) | $50.86^{\pm 1.2}$ | $78.19^{\pm 0.9}$ (↑**27.33**) |
| FL+A-GEM | $24.43^{\pm 2.1}$ | $23.29^{\pm 3.8}$ (↓**1.14**) | $8.62^{\pm 0.1}$ | $19.58^{\pm 1.2}$ (↑**10.96**) | $63.02^{\pm 0.6}$ | $76.23^{\pm 0.6}$ (↑**13.21**) |
| FL+DER | $17.89^{\pm 1.3}$ | $46.17^{\pm 3.0}$ (↑**28.28**) | $11.53^{\pm 0.5}$ | $26.64^{\pm 2.8}$ (↑**15.11**) | $57.00^{\pm 1.4}$ | $69.42^{\pm 1.0}$ (↑**12.42**) |

Table 8: Average accuracy $\text{Acc}_T$ (%) on S-TinyImageNet

| Methods | S-TinyImageNet | |
|---|---|---|
| | *Class-IL* | *Task-IL* |
| FL | 6.48 | 32.41 |
| FL+A-GEM | 6.58 (↑0.10) | 39.66 (↑7.25) |
| FL+DER | 6.36 (↓0.12) | 32.53 (↑0.12) |
| FL+Ours | **7.95** (↑**1.47**) | **45.97** (↑**13.56**) |
| FL+A-GEM+Ours | **10.20** (↑**3.72**) | **50.27** (↑**17.86**) |

### 5.1.10 Effect of Out-of-Distribution (OOD) Datasets

To further validate Fed-A-GEM effectiveness in handling distribution shifts, we conducted experiments on two challenging OOD benchmarks: PACS (Li et al., 2017) and Office-Caltech-10 (Wang et al., 2018). These datasets are particularly relevant for evaluating continual learning methods as they introduce natural domain shifts that models must adapt to while preserving knowledge. The PACS dataset consists of images across four distinct domains (Photo, Art, Cartoon, Sketch), presenting significant visual style variations. Office-Caltech-10 contains images of office objects from different sources (Amazon, DSLR, Webcam, Caltech), introducing real-world domain shifts. We have maintained consistent experimental conditions with our main experiments: ResNet18 (pretrained on ImageNet) as the backbone, 10 clients with non-IID data distribution, and a memory buffer size of 200 per client. The results demonstrate that Fed-A-GEM significantly improves performance on OOD data, with particularly strong gains on the more challenging PACS dataset (+30.42%). These results complement our main findings and demonstrate Fed-A-GEM effectiveness in challenging real-world scenarios where distribution shifts are common.

Table 9: Average accuracy $\text{Acc}_T$ (%) on Out-of-Distribution datasets

| Methods | Out-of-Distribution Datasets | |
|---|---|---|
| | *Office-Caltech-10* | *PACS* |
| FL | 90.47±1.2 | 40.30±2.1 |
| FL+Fed-A-GEM | **93.10±0.8** (↑**2.63**) | **70.72±1.8** (↑**30.42**) |

## 5.2 Text Classification

In addition to image classification, we also extended the evaluation of our method on text classification task (Mehta et al., 2023). For this purpose, we utilized the YahooQA (Zhang et al., 2015) dataset which comprises texts (questions and answers), and user-generated labels representing 10 different topics. Similar to the approach taken with the CIFAR10 dataset, we partitioned the YahooQA dataset into 5 tasks, where each task consisted of two distinct classes. Within each task, we used LDA to partition data across 10 clients in a non-IID manner. To conduct the experiment, we employed a pretrained DistilBERT (Sanh et al., 2019) with

linear classification layer. We freeze the DistilBERT model and only fine-tune the additional linear layer. The results of this experiment can be found in Table 10. We can observe that Fed-A-GEM consistently enhances the accuracy ($\text{Acc}_T$) over baselines, particularly in class-IL scenarios. For example, FL+A-GEM+Fed-A-GEM achieves 47.02% accuracy, while the baseline FL achieves 17.86%.

Table 10: Average classification accuracy $\text{Acc}_T$ (%) on split-YahooQA dataset.

| | sequential-YahooQA (*Class-IL*) | | sequential-YahooQA (*Task-IL*) | |
| Method | w/o Fed-A-GEM | w/ Fed-A-GEM | w/o Fed-A-GEM | w/ Fed-A-GEM |
|---|---|---|---|---|
| FL | $17.86^{\pm0.6}$ | $30.67^{\pm4.4}(\uparrow\mathbf{12.81})$ | $80.87^{\pm1.2}$ | $88.04^{\pm1.4}(\uparrow\mathbf{7.17})$ |
| FL+A-GEM | $20.86^{\pm0.3}$ | $47.02^{\pm1.9}(\uparrow\mathbf{26.16})$ | $87.29^{\pm1.3}$ | $90.20^{\pm0.2}(\uparrow\mathbf{2.91})$ |
| FL+DER | $43.64^{\pm2.1}$ | $54.28^{\pm1.3}(\uparrow\mathbf{10.64})$ | $89.57^{\pm0.2}$ | $90.48^{\pm0.2}(\uparrow\mathbf{0.91})$ |

### 5.3 Ablations on Algorithm Design

We have performed ablation studies on our algorithm, which consists of two main components: the gradient refinement algorithm and the buffer updating algorithm. These experiments help in understanding the individual contributions of each component to the overall performance of our system.

#### 5.3.1 Gradient Refinement

First, we considered different ways of refining the gradient $g$, given the reference gradient $g_{\text{ref}}$. We define the refined gradient $\tilde{g}$ as follows:

- **Average**: The gradient $\tilde{g}$ is the arithmetic mean of $g$ and $g_{\text{ref}}$, expressed as $\tilde{g} = \frac{g+g_{\text{ref}}}{2}$.

- **Rotate**: The gradient $g$ is rotated towards $g_{\text{ref}}$, maintaining its original magnitude. This can be represented as:

$$\tilde{g} = \|g\| \frac{g + g_{\text{ref}}}{\|g + g_{\text{ref}}\|}$$

  where $\|g\|$ denotes the magnitude of $g$.

- **Project**: $g$ is projected onto a space orthogonal to $g_{\text{ref}}$.

- **Project & Scale**: This method extends the Project method by scaling the resultant vector to match the original magnitude of $g$.

Our Fed-A-GEM applies "Project" method only when the angle between $g$ and $g_{\text{ref}}$ is larger than 90 degree, i.e., when the reference gradient $g_{\text{ref}}$ (measured for the previous tasks) and the gradient $g$ (measured for the current task) conflicts to each other. Our intuition for such choice is, it is better to manipulate $g$ if the direction favorable for current task is conflicting with the direction favorable for previous tasks. To support that this choice is meaningful, we compared two ways of deciding when we manipulate the gradients:

- **Conditional Refinement** ($> 90°$): Gradient $g$ is updated only when the angle between $g$ and $g_{\text{ref}}$ is greater than 90 degrees.

- **Unconditional Refinement** (Always): Gradient $g$ is always updated irrespective of the angle.

The performance of these strategies is demonstrated in Table 11, for sequential-CIFAR100 dataset. The results reveal that our Fed-A-GEM (denoted by Project ($> 90$) in the table) far outperforms all other combinations, showing that our design (doing projection for conflicting case only) is the right choice. Investigating each component (Project and ($> 90$)) independently, we can observe that choosing "Project" outperforms "Average", "Rotate" and "Project & Scale" in most cases, and choosing ($> 90$) outperforms "Always" in all cases.

Table 11: Effect of gradient refinement methods on the accuracy (%) of FedGP on sequential-CIFAR100

| Method | Class-IL | Task-IL |
|---|---|---|
| FL | $8.76^{\pm0.1}$ | $47.74^{\pm1.2}$ |
| Average (Always) | $7.26^{\pm1.95}$ | $35.96^{\pm3.23}$ |
| Average ($> 90$) | $7.79^{\pm0.65}$ | $36.57^{\pm1.55}$ |
| Rotate (Always) | $7.59^{\pm0.89}$ | $36.15^{\pm2.83}$ |
| Rotate ($> 90$) | $8.41^{\pm0.78}$ | $38.97^{\pm1.83}$ |
| Project & Scale (Always) | $8.77^{\pm0.09}$ | $32.96^{\pm1.10}$ |
| Project & Scale ($> 90$) | $12.30^{\pm0.65}$ | $73.61^{\pm0.75}$ |
| Project (Always) | $8.90^{\pm0.08}$ | $34.00^{\pm1.98}$ |
| Project ($> 90$), **ours** | $\mathbf{17.08^{\pm1.8}}$ | $\mathbf{74.71^{\pm0.9}}$ |

We also tested whether doing the projection is helpful in all cases when $\text{angle}(g, g_{\text{ref}}) > 90$. We considered applying the projection for $p\%$ of the cases having $\text{angle}(g, g_{\text{ref}}) > 90$, for $p = 10, 50, 80$ and $100$. Note that $p = 100\%$ case reduces to our Fed-A-GEM. Table 12 shows the effect of projection rate $p\%$ on the accuracy, tested on sequential-CIFAR100 dataset. In both class-IL and task-IL settings, increasing $p$ always improves the accuracy of the Fed-A-GEM method. This supports that the projection used in our method is suitable for the continual federated learning setup.

Table 12: Effect of projection rate $p\%$ on the accuracy (%) of Fed-A-GEM, tested on sequential-CIFAR100

| Method | Class-IL | Task-IL |
|---|---|---|
| FL, $p = 0\%$ | $8.76^{\pm0.1}$ | $47.74^{\pm1.2}$ |
| Fed-A-GEM, $p = 10\%$ | $8.82^{\pm0.07}$ | $54.90^{\pm1.61}$ |
| Fed-A-GEM, $p = 50\%$ | $8.91^{\pm0.07}$ | $67.89^{\pm0.67}$ |
| Fed-A-GEM, $p = 80\%$ | $10.36^{\pm0.42}$ | $72.73^{\pm0.74}$ |
| Fed-A-GEM, $p = 100\%$ (**ours**) | $\mathbf{17.08^{\pm1.8}}$ | $\mathbf{74.71^{\pm0.9}}$ |

### 5.3.2 Buffer Updating

In Table 13, we compared three different buffer updating algorithms:

- **Sliding Window Sampling**: This method replaces the earliest data point in the buffer when new data arrives

- **Random Sampling**: It randomly replaces a data point in the buffer with incoming new data

- **Reservoir Sampling (Vitter, 1985) (Used in Ours)**: We employ it for a buffer of size $|\mathcal{M}^k|$ and $n$ total number of observed samples up to now, which operates as follows:

  - When $n \leq |\mathcal{M}^k|$, we simply add the current sample to the buffer.
  - When $n > |\mathcal{M}^k|$, with probability $\frac{|\mathcal{M}^k|}{n}$ we replace a sample in buffer with the current sample.

This method ensures that each of the $n$ samples has an equal probability of being included in the buffer, crucial for maintaining uniform sample representation from each task throughout the continual learning process. The effectiveness of using Reservoir Sampling in our method is validated in Table 13, where it outperforms other methods.

Table 13: Effect of buffer updating algorithms on the accuracy (%) of Fed-A-GEM, tested on S-CIFAR100

| Method | *Class-IL* | *Task-IL* |
|---|---|---|
| FL | $8.76^{\pm 0.1}$ | $47.74^{\pm 1.2}$ |
| Sliding Window Sampling | $8.82^{\pm 0.15}$ | $46.16^{\pm 2.38}$ |
| Random Sampling | $9.72^{\pm 0.10}$ | $54.82^{\pm 1.58}$ |
| Reservoir Sampling **(used in ours)** | $\mathbf{17.08^{\pm 1.8}}$ | $\mathbf{74.71^{\pm 0.9}}$ |

We also present the pseudocode for the `ReservoirSampling` algorithm in Algorithm 4. Reservoir sampling ensures each sample has an equal probability of being included in the buffer. The probability of a sample being contained in the buffer is $\frac{|\mathcal{M}^k|}{n}$. This can be shown by induction, assuming the statement is true for $n-1$ samples and showing it holds when one additional sample is observed. The probability of a sample contained in the buffer can be computed as $\frac{|\mathcal{M}^k|}{n-1} \times (1 - \frac{|\mathcal{M}^k|}{n} \times \frac{1}{|\mathcal{M}^k|}) = \frac{|\mathcal{M}^k|}{n}$, where

- $\frac{|\mathcal{M}^k|}{n-1}$ is the probability that a sample is initially in the buffer.

- $(1 - \frac{|\mathcal{M}^k|}{n} \times \frac{1}{|\mathcal{M}^k|})$ is the probability that a sample is not displaced from the buffer.

- $\frac{|\mathcal{M}^k|}{n} \times \frac{1}{|\mathcal{M}^k|}$ is the probability that a sample is displaced from the buffer.

In conclusion, the reservoir sampling method used in Fed-A-GEM allows us to have balanced sample distribution across different tasks, thus allowing us to mitigate catastrophic forgetting and to improve the accuracy in the continual federated learning setting. Additionally, we recognize the potential of biased reservoir sampling (Aggarwal, 2006), a variant of reservoir sampling that employs exponential bias functions to prioritize recent data. This technique is particularly advantageous for data streams where recent observations are more significant. We believe that integrating such approaches could align with and further complement our method.

## 6   Conclusion

In this paper, we present Fed-A-GEM, a simple yet highly effective method of using buffer data for mitigating the catastrophic forgetting issues in CFL. Our approach leverages a buffer-based gradient projection strategy that integrates seamlessly with existing CFL techniques to enhance their performance across various tasks and settings. Through extensive experiments on benchmark datasets such as rotated-MNIST, permuted-MNIST, sequential-CIFAR10, sequential-CIFAR100, and sequential-YahooQA, we demonstrate that Fed-A-GEM consistently improves accuracy and reduces forgetting compared to baseline methods. Notably, our method achieves these improvements without increasing the communication overhead between the server and clients.

Despite these promising results, our work has certain limitations. First, while Fed-A-GEM effectively mitigates forgetting, it requires maintaining a buffer of past samples, which might not be feasible for all devices. Second, the assumption of secure aggregation, while essential for preserving privacy, adds computational overhead that may affect scalability in extremely large federated networks. There are several directions for future research. One potential avenue is to explore more efficient buffer management strategies that further reduce storage requirements while maintaining performance. Another interesting direction is to integrate advanced privacy-preserving techniques, such as differential privacy or homomorphic encryption, with our approach to enhance the security and privacy guarantees in sensitive applications.

## Acknowledgments

Authors are supported in part by the following: US National Science Foundation awards — CNS-2112562, CNS-2107060, CNS-2213688, CNS-2312716, and the US Department of Commerce award 70NANB21H043. Jy-yong Sohn is supported by the National Research Foundation of Korea (NRF) grant funded by the Korea

government (Ministry of Science and ICT; MSIT) (RS-2024-00345351, RS-2024-00408003), the IITP(Institute of Information & Coummunications Technology Planning & Evaluation)-ICAN(ICT Challenge and Advanced Network of HRD)(RS-2023-00259934) grant funded by the Korea government (MSIT), and the Yonsei University Research Fund (2024-22-0124). Kangwook Lee is supported by NSF CAREER Award CCF2339978, an Amazon Research Award, and a grant from FuriosaAI.

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

## A  Incremental Learning Settings in Continual Learning

We describe three types of incremental learning (IL) settings in Continual Learning (CL) with examples.

**Task-IL (Task-Incremental Learning):**  In this setting, each task is explicitly labeled, and the model knows which task it is performing during testing. For example, a model learns to classify: Task 1: Dogs vs. Cats, and Task 2: Cars vs. Buses. At test time, the model is informed whether "this is a dog/cat classification task" or "this is a vehicle classification task." This scenario is considered simpler because the model can utilize task-specific parameters or heads, given that it knows the task it is performing.

**Domain-IL (Domain-Incremental Learning):**  The classes remain the same across tasks, but the input distribution changes. For instance, consider the task of classifying digits (0-9) across different domains: Task 1: Handwritten digits (MNIST) and Task 2: Street View House Numbers (SVHN). While the classes (0-9) remain constant, the visual characteristics (domain) vary. The model must learn to recognize the same classes under differing conditions and styles.

**Class-IL (Class-Incremental Learning):**  In this setting, new classes are introduced with each task, and the model must learn to distinguish among all classes encountered so far. For example, consider an animal classifier learning: Task 1: Dog and Cat, Task 2: Bird and Fish, and Task 3: Horse and Cow. At test time, after learning Task 3, the model is required to correctly classify any of the six animals without being informed which task the input belongs to. This is often regarded as the most challenging setting.

## B  Reservoir Sampling

In the following, we provide the buffer updating algorithm for the Reservoir Sampling strategy.

---

**Algorithm 4** ReservoirSampling$(\mathcal{M}^k, \{(x_i, y_i)\}_{i=1}^{\beta}, n)$

---

**Input:** local buffer $\mathcal{M}^k$, incoming data $\{(x_i, y_i)\}_{i=1}^{\beta}$ and the number of previously observed samples $n$
**for** $i = 1$ to $\beta$ **do**
    $n \leftarrow n + 1$
    **if** $n \leq |\mathcal{M}^k|$ **then**
        Add data $(x_i, y_i)$ into local buffer $\mathcal{M}^k$
    **else**
        $j \leftarrow \text{Uniform}\{1, 2, \cdots, n\}$
        **if** $j \leq |\mathcal{M}^k|$ **then**
            $\mathcal{M}^k[j] \leftarrow (x_i, y_i)$
        **end if**
    **end if**
**end for**
Return $\mathcal{M}^k$, the updated local buffer

---

# C   Supplementary Results

In this section, we furnish additional experimental outcomes that serve to further bolster the findings of our primary investigation.

## C.1   Forgetting analysis across datasets

We present the complementary information to Table 1 in Table 14, illustrating the extent of $\text{Fgt}_T$ observed across multiple benchmark datasets. Our method exhibits exceptional effectiveness in mitigating forgetting. Remarkably, it demonstrates consistent performance across all datasets and baselines, making it a versatile solution.

Table 14: Average forgetting $\text{Fgt}_T$ (%) (lower is better) on benchmark datasets at the final task $T$.

| | **rotated-MNIST** (*Domain-IL*) | | **sequential-CIFAR10** (*Class-IL*) | | **sequential-CIFAR10** (*Task-IL*) | |
|---|---|---|---|---|---|---|
| Method | w/o Fed-A-GEM | w/ Fed-A-GEM | w/o Fed-A-GEM | w/ Fed-A-GEM | w/o Fed-A-GEM | w/ Fed-A-GEM |
| FL | $25.98^{\pm3.2}$ | $11.66^{\pm2.7}$ ($\downarrow$**14.32**) | $80.69^{\pm3.6}$ | $78.62^{\pm4.3}$ ($\downarrow$**2.07**) | $15.37^{\pm4.8}$ | $4.49^{\pm1.9}$ ($\downarrow$**10.88**) |
| FedCurv | $25.80^{\pm2.4}$ | $11.18^{\pm2.7}$ ($\downarrow$**14.62**) | $80.90^{\pm6.6}$ | $79.85^{\pm3.9}$ ($\downarrow$**1.05**) | $19.37^{\pm4.8}$ | $4.77^{\pm1.6}$ ($\downarrow$**14.60**) |
| FedProx | $25.74^{\pm3.1}$ | $11.76^{\pm2.9}$ ($\downarrow$**13.98**) | $84.35^{\pm2.4}$ | $80.24^{\pm2.5}$ ($\downarrow$**4.11**) | $18.24^{\pm4.9}$ | $4.17^{\pm1.0}$ ($\downarrow$**14.07**) |
| FL+A-GEM | $26.30^{\pm5.7}$ | $15.18^{\pm2.4}$ ($\downarrow$**11.12**) | $82.18^{\pm6.6}$ | $80.38^{\pm2.5}$ ($\downarrow$**1.80**) | $10.00^{\pm3.0}$ | $4.15^{\pm0.7}$ ($\downarrow$**5.85**) |
| FL+DER | $21.42^{\pm4.0}$ | $5.51^{\pm1.2}$ ($\downarrow$**15.91**) | $60.98^{\pm14.6}$ | $47.88^{\pm7.2}$ ($\downarrow$**13.10**) | $6.34^{\pm4.9}$ | $2.73^{\pm1.3}$ ($\downarrow$**3.61**) |
| | **permuted-MNIST** (*Domain-IL*) | | **sequential-CIFAR100** (*Class-IL*) | | **sequential-CIFAR100** (*Task-IL*) | |
| FL | $43.47^{\pm5.3}$ | $21.40^{\pm4.9}$ ($\downarrow$**22.07**) | $77.69^{\pm0.5}$ | $67.02^{\pm2.3}$ ($\downarrow$**10.67**) | $34.38^{\pm1.6}$ | $5.39^{\pm0.8}$ ($\downarrow$**28.99**) |
| FedCurv | $42.88^{\pm5.0}$ | $22.85^{\pm3.5}$ ($\downarrow$**20.03**) | $78.40^{\pm0.9}$ | $67.75^{\pm0.8}$ ($\downarrow$**10.65**) | $33.71^{\pm2.2}$ | $5.86^{\pm0.7}$ ($\downarrow$**27.85**) |
| FedProx | $42.59^{\pm5.6}$ | $20.77^{\pm5.6}$ ($\downarrow$**21.82**) | $77.35^{\pm0.4}$ | $66.81^{\pm2.2}$ ($\downarrow$**10.54**) | $34.79^{\pm3.6}$ | $5.69^{\pm0.9}$ ($\downarrow$**29.10**) |
| FL+A-GEM | $35.61^{\pm5.3}$ | $24.05^{\pm2.4}$ ($\downarrow$**11.56**) | $77.97^{\pm0.7}$ | $63.99^{\pm2.0}$ ($\downarrow$**13.98**) | $16.92^{\pm1.1}$ | $5.16^{\pm0.5}$ ($\downarrow$**11.76**) |
| FL+DER | $45.33^{\pm5.0}$ | $34.71^{\pm5.0}$ ($\downarrow$**10.62**) | $69.37^{\pm1.7}$ | $53.84^{\pm6.7}$ ($\downarrow$**15.53**) | $22.43^{\pm0.7}$ | $14.16^{\pm1.7}$ ($\downarrow$**8.27**) |

In line with the presentation of forgetting in Table 14, we present the forgetting analysis when the number of clients is set to 20 in Table 15. Notably, our method exhibits consistent and impressive performance across varying numbers of users. It consistently proves its effectiveness regardless of the specific user count, showcasing its robustness and reliability.

Table 15: The $\text{Fgt}_T$ (%) (lower is better) performance measured when we have $K = 20$ users.

| | rotated-MNIST (*Domain-IL*) | | sequential-CIFAR10 (*Class-IL*) | | sequential-CIFAR10 (*Task-IL*) | |
|---|---|---|---|---|---|---|
| Method | w/o Fed-A-GEM | w/ Fed-A-GEM | w/o Fed-A-GEM | w/ Fed-A-GEM | w/o Fed-A-GEM | w/ Fed-A-GEM |
| FL | $31.00^{\pm 9.5}$ | $13.45^{\pm 3.6}$ ($\downarrow\mathbf{17.55}$) | $82.62^{\pm 3.1}$ | $73.39^{\pm 4.5}$ ($\downarrow\mathbf{9.23}$) | $17.93^{\pm 2.7}$ | $6.14^{\pm 4.9}$ ($\downarrow\mathbf{11.79}$) |
| FedCurv | $30.73^{\pm 9.3}$ | $12.97^{\pm 3.8}$ ($\downarrow\mathbf{17.76}$) | $79.55^{\pm 3.8}$ | $75.38^{\pm 5.3}$ ($\downarrow\mathbf{4.17}$) | $18.19^{\pm 3.0}$ | $9.14^{\pm 3.1}$ ($\downarrow\mathbf{9.05}$) |
| FedProx | $31.04^{\pm 9.7}$ | $13.31^{\pm 3.4}$ ($\downarrow\mathbf{17.73}$) | $82.94^{\pm 1.1}$ | $78.67^{\pm 4.2}$ ($\downarrow\mathbf{4.27}$) | $20.60^{\pm 2.6}$ | $8.52^{\pm 3.0}$ ($\downarrow\mathbf{12.08}$) |
| FL+A-GEM | $25.22^{\pm 8.8}$ | $11.02^{\pm 3.0}$ ($\downarrow\mathbf{14.20}$) | $82.39^{\pm 2.4}$ | $80.25^{\pm 4.1}$ ($\downarrow\mathbf{2.14}$) | $12.29^{\pm 2.2}$ | $4.00^{\pm 2.4}$ ($\downarrow\mathbf{8.29}$) |
| FL+DER | $28.93^{\pm 6.6}$ | $5.18^{\pm 1.1}$ ($\downarrow\mathbf{23.75}$) | $55.10^{\pm 9.8}$ | $60.90^{\pm 3.8}$ ($\uparrow\mathbf{5.80}$) | $3.20^{\pm 1.6}$ | $2.71^{\pm 1.7}$ ($\downarrow\mathbf{0.49}$) |
| | permuted-MNIST (*Domain-IL*) | | sequential-CIFAR100 (*Class-IL*) | | sequential-CIFAR100 (*Task-IL*) | |
| FL | $24.27^{\pm 5.2}$ | $8.67^{\pm 7.0}$ ($\downarrow\mathbf{15.60}$) | $73.05^{\pm 0.5}$ | $62.71^{\pm 0.9}$ ($\downarrow\mathbf{10.34}$) | $27.07^{\pm 1.7}$ | $2.48^{\pm 0.7}$ ($\downarrow\mathbf{24.59}$) |
| FedCurv | $24.02^{\pm 5.4}$ | $8.10^{\pm 5.4}$ ($\downarrow\mathbf{15.92}$) | $80.07^{\pm 0.5}$ | $68.58^{\pm 1.1}$ ($\downarrow\mathbf{11.49}$) | $34.63^{\pm 1.7}$ | $3.48^{\pm 0.6}$ ($\downarrow\mathbf{31.15}$) |
| FedProx | $23.01^{\pm 5.7}$ | $5.93^{\pm 5.1}$ ($\downarrow\mathbf{17.08}$) | $79.46^{\pm 0.5}$ | $68.40^{\pm 0.9}$ ($\downarrow\mathbf{11.06}$) | $32.82^{\pm 1.4}$ | $4.13^{\pm 0.7}$ ($\downarrow\mathbf{28.69}$) |
| FL+A-GEM | $22.12^{\pm 4.9}$ | $9.45^{\pm 5.4}$ ($\downarrow\mathbf{12.67}$) | $72.97^{\pm 1.1}$ | $60.27^{\pm 1.3}$ ($\downarrow\mathbf{12.70}$) | $12.54^{\pm 1.3}$ | $2.66^{\pm 0.2}$ ($\downarrow\mathbf{9.88}$) |
| FL+DER | $32.26^{\pm 1.1}$ | $27.30^{\pm 4.2}$ ($\downarrow\mathbf{4.96}$) | $67.07^{\pm 0.8}$ | $47.74^{\pm 3.8}$ ($\downarrow\mathbf{19.33}$) | $19.78^{\pm 1.7}$ | $8.67^{\pm 1.4}$ ($\downarrow\mathbf{11.11}$) |

## C.2 Progressive performance of Fed-A-GEM across tasks

Fig. 5 depicts the average accuracy $\text{Acc}_t$ measured at task $t = 1, 2, \cdots, 10$ and the average forgetting $\text{Fgt}_t$ measured at task $t = 2, 3, \cdots, 10$. The accuracy of FedAvg rapidly drops as different tasks are given to the model, as expected. FedCurv and FedProx perform similarly to FedAvg, while A-GEM and DER partially alleviate forgetting, resulting in higher accuracies and reduced forgetting compared to FedAvg. Combining these baselines with Fed-A-GEM lead to significant performance improvements, which allows the solid lines in the accuracy plot consistently remain at the top. For example, for the experiment on task-IL for sequential-CIFAR100, the accuracy measured at task 5 (denoted by $\text{Acc}_5$) is 55.37% for FedProx, while 71.12% for FedProx+Fed-A-GEM. These results demonstrate that Fed-A-GEM effectively mitigates forgetting and enhances existing methods in CFL.

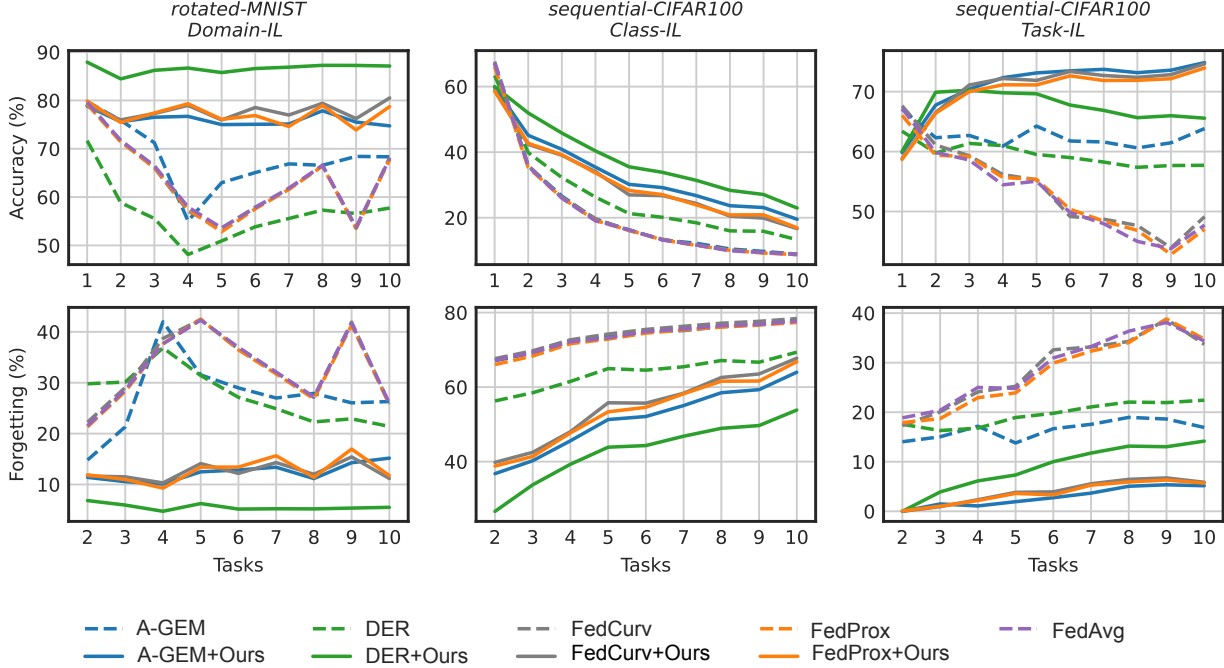

Figure 5: Evaluating accuracy ($\uparrow$) and forgetting ($\downarrow$) in multiple datasets with and without Fed-A-GEM using a buffer size of 200. The solid lines indicate the results obtained with our method, while the dotted lines represent the results obtained without our method. The results show a significant improvement in accuracy as well as reduced forgetting for all settings.

### C.3 Random sampling

We implement a more realistic federated learning environment by applying uniform sampling techniques to randomly select the participating clients in each round. We conduct experiments on CIFAR100. A total of 50 clients is set up, and during each communication, only a random 50% of the clients participate in training. As can be seen, even in such a scenario, where our algorithm cannot update the reference gradient using the local buffer from all clients, there is still an improvement in performance using our algorithm.

Table 16: Average accuracy $\text{Acc}_T$ (%) with 50 clients and 50% client sampling rate, for sequential-CIFAR100

| Method | Class-IL | Task-IL |
|---|---|---|
| FL | $7.46 \pm 0.08$ | $43.85 \pm 1.33$ |
| FL+Fed-A-GEM | $9.34 \pm 0.31\,(\uparrow 1.88)$ | $65.76 \pm 0.48\,(\uparrow 21.91)$ |

Moreover, inspired by the client incremental setup described in the GLFC paper, we simulated a dynamic environment where new clients and new classes are gradually introduced, assuming a non-IID data distribution among clients. Starting with 30 local clients in the CIFAR-100 setting, we introduced 10 additional new local clients with each incremental task. At each global round, 10 clients were randomly selected to achieve partial client participation. The results are as follows.

Table 17: Average accuracy $\text{Acc}_T$ (%) under client incremental scenario

| Methods | Class-IL | Task-IL |
|---|---|---|
| FL | $9.31^{\pm 0.0}$ | $49.81^{\pm 1.9}$ |
| FL+A-GEM | $9.38^{\pm 0.1}$ | $56.81^{\pm 1.4}$ |
| FL+Ours | $10.22^{\pm 0.3}$ | $70.63^{\pm 1.0}$ |
| FL+A-GEM+Ours | $\mathbf{10.82^{\pm 0.7}}$ | $\mathbf{72.88^{\pm 0.6}}$ |

As can be seen, our method (FL+Fed-A-GEM) demonstrates significant improvements in Task-IL accuracy compared to the baseline FL, even in this more complex scenario involving the introduction of new clients. The parameter details of this experimental setup are shown in Table 18.

Table 18: The parameter detail of introducing new client setting that differs from our main setting.

| Parameter | Value |
|---|---|
| learning rate | 2.0 |
| buffer size | 2000 |
| number of local training epoch | 20 |
| number of tasks | 10 |
| local batch size | 128 |
| communication per task | 200 |
| number of clients | 120 |
| weight decay | 0.00001 |
| client participation rate | 0.512 |

### C.4 Performance on the current task

Balancing the retention of old tasks and the learning of new ones is a common challenge in continual learning. It can be difficult to determine the best approach, especially when two tasks are significantly different. This is a challenge faced by many methods in continual learning.

We provided additional experimental results on the performance measured for the current task. The below Fig. 6 shows the Class-IL accuracy of Fed-A-GEM (with buffer size 200) and FL for sequential-CIFAR100,

where the total number of tasks is set to 10. During the continual learning process, we measured the accuracy of each model for the current task. One can confirm that using Fed-A-GEM does not hurt the current task accuracy, compared with FL. Note that this shows that Fed-A-GEM does not impair the performance of the current task, while also alleviating the forgetting in upcoming rounds.

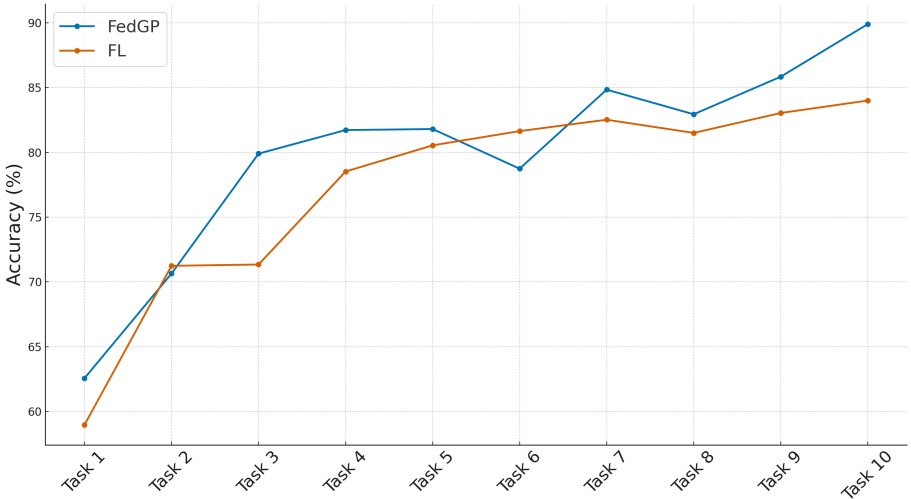

Figure 6: Class-IL Accuracy (%) of current task for FL and FedGP on the sequential-CIFAR100

### C.5 Backward and forward transfer metrics

1. **Backward Transfer (BWT):**
   Backward Transfer measures the influence that learning a new task has on the performance of previously learned tasks. After the model finishes learning all $T$ tasks, BWT is defined as:

$$\text{BWT} = \frac{1}{T-1} \sum_{i=1}^{T-1} (a_{T,i} - a_{i,i})$$

   where:
   - $a_{T,i}$ is the accuracy of the global model on task $i$ after training up to task $T$.
   - $a_{i,i}$ is the accuracy of the global model on task $i$ after training up to task $i$.

2. **Forward Transfer (FWT):**
   Forward Transfer measures the influence that learning a task has on the performance of future tasks. After the model finishes learning all $T$ tasks, FWT is defined as:

$$\text{FWT} = \frac{1}{T-1} \sum_{i=2}^{T} (a_{i-1,i} - \bar{b}_i)$$

   where:
   - $a_{i-1,i}$ is the accuracy of the global model on task $i$ before learning task $i$.
   - $\bar{b}_i$ is the baseline accuracy of task $i$ at random initialization.

Our method outperforms FedAvg (FL) in both Backward and Forward Transfer metrics across the sequential-CIFAR10 and sequential-CIFAR100 datasets, as shown in the Table 19.

Table 19: Backward and Forward Transfer (↑) Results for sequential-CIFAR100 and sequential-CIFAR10

| Metric | Dataset | Methods | *Class-IL* | *Task-IL* |
|--------|---------|---------|-----------|----------|
| Backward | CIFAR100 | FL | $-78.11$ | $-36.52$ |
| Backward | CIFAR100 | FL+Fed-A-GEM | $-72.24\,(\uparrow 5.87)$ | $-3.68\,(\uparrow 32.84)$ |
| Backward | CIFAR10 | FL | $-78.78$ | $-14.48$ |
| Backward | CIFAR10 | FL+Fed-A-GEM | $-78.55\,(\uparrow 0.23)$ | $-0.60\,(\uparrow 13.88)$ |
| Forward | CIFAR100 | FL | $16.98$ | $16.98$ |
| Forward | CIFAR100 | FL+Fed-A-GEM | $17.16\,(\uparrow 0.18)$ | $17.48\,(\uparrow 0.50)$ |
| Forward | CIFAR10 | FL | $12.75$ | $12.74$ |
| Forward | CIFAR10 | FL+Fed-A-GEM | $12.98\,(\uparrow 0.23)$ | $12.99\,(\uparrow 0.25)$ |

## C.6 Effect of different curriculum.

We evaluate how the performance of Fed-A-GEM changes when we shuffle the order of tasks in the continual learning. We randomly shuffle the sequential-CIFAR100 task order and label them as curriculum 1 to 4, as shown in the Table 20. Regardless of the different curriculum, FL+Fed-A-GEM outperforms FedAvg.

Table 20: Average accuracy $\text{Acc}_T$ (%) across randomized curriculum in sequential-CIFAR100.

| Curriculum | Methods | *Class-IL* | *Task-IL* |
|-----------|---------|-----------|----------|
| 1 | FL | 8.15 | 46.25 |
| 1 | FL+Fed-A-GEM | $12.10\,(\uparrow 3.95)$ | $72.69\,(\uparrow 26.44)$ |
| 2 | FL | 8.46 | 47.56 |
| 2 | FL+Fed-A-GEM | $14.37\,(\uparrow 5.91)$ | $73.19\,(\uparrow 25.63)$ |
| 3 | FL | 8.82 | 45.04 |
| 3 | FL+Fed-A-GEM | $12.58\,(\uparrow 3.76)$ | $74.71\,(\uparrow 29.67)$ |
| 4 | FL | 7.87 | 43.87 |
| 4 | FL+Fed-A-GEM | $14.85\,(\uparrow 6.98)$ | $73.74\,(\uparrow 29.87)$ |

## C.7 Additional hyperparameters for specific methods

In addition to the hyperparameters discussed in the main paper, additional method-specific hyperparameters are outlined in Table 21.

Table 21: Additional hyperparameters for specific methods.

| Method | Parameter | Values |
|--------|-----------|--------|
| FL+DER | Regularization Coefficient | sequential-CIFAR10 (0.3), Others (1) |
| FL+L2P | Communication Round $R$ | rotated-MNIST (5), permuted-MNIST (1), sequential-CIFAR10 (20), sequential-CIFAR100 (20) |
| CFeD | Surrogate Dataset | sequential-CIFAR10 (CIFAR100), sequential-CIFAR100 (CIFAR10) |
| | Note: No server distillation included. | |

# D Object Detection

Here we test Fed-A-GEM on realistic streaming data (Dai et al., 2023) which leverage two open source tools, an urban driving simulator (CARLA (Dosovitskiy et al., 2017)) and a FL framework (OpenFL (Reina et al., 2021)). As shown in Fig. 7a, CARLA provides OpenFL with a real-time collection of continuous streaming vehicle camera output data and automatic annotation about object detection. This streaming data capture the spatio-temporal dynamics of data generated from real-world applications. After loading data of vehicles from CARLA, OpenFL performs collaborative training over multiple clients.

We evaluate the solutions to the forgetting problem by spawning two vehicles in a virtual town. During the training of the tinyYOLO (Redmon & Farhadi, 2017) object detection model, we use a custom loss that combines classification, detection and confidence losses. In order to quantify the quality of the incremental

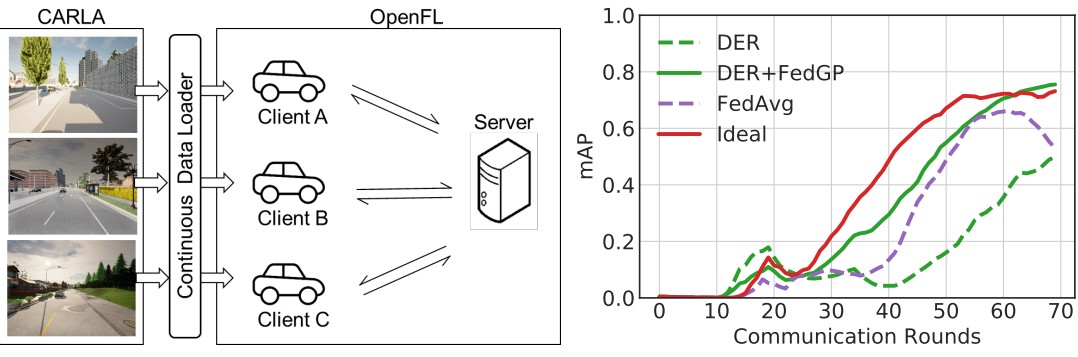

(a) Framework for automotive data evaluation.  (b) Object detection performance comparison.

Figure 7: (a) The data loader continuously supplies data from CARLA camera outputs to individual FL clients. Each client trains on its local data and updates its buffer to retain old knowledge. (b) The result shows the object detection performance comparison between Ideal, FedAvg, DER, and DER+Fed-A-GEM on a realistic CARLA dataset.

| **Algorithm 5** DER `ClientUpdate` at client $k$ | **Algorithm 6** A-GEM `ClientUpdate` at client $k$ |
|---|---|
| **Input:** Task index $t$, model $w$, buffer gradient $g_{\text{ref}}$, batch size $\beta$, regularization coefficient $\lambda$, and learning rate $\alpha$ 
 Load the dataset $\mathcal{D}_t^k$ and local buffer $\mathcal{M}^k$ 
 Initialize $n = 0$ at the first task 
 **for** each batch $\{(x_i, y_i)\}_{i=1}^{\beta}$ in $\mathcal{D}_t^k$ **do** 
    Let $X = \{x_i\}_{i=1}^{\beta}$ and $Y = \{y_i\}_{i=1}^{\beta}$ 
    $Z \leftarrow h(X; w)$ 
    where $f(X; w) := \sigma\left(h(X; w)\right)$ 
    Sample $(X', Z', Y')$ from $\mathcal{M}^k$ 
    $\ell_{\text{reg}} \leftarrow \lambda \| Z' - h(X'; w) \|_2^2$ 
    $g = \nabla_w \left[ \ell(Y, f(X; w)) + \ell_{\text{reg}} \right]$ 
    $\tilde{g} \leftarrow g - \text{proj}_{g_{\text{ref}}} g \cdot \mathbf{1}(g_{\text{ref}}^\top g \leq 0)$ 
    $w \leftarrow w - \alpha \tilde{g}$ 
    `ReservoirSampling`$(\mathcal{M}^k, (X, Z, Y), n)$ 
    $n \leftarrow n + \beta$ 
 **end for** 
 Return $w$ | **Input:** Task index $t$, model $w$, buffer gradient $g_{\text{ref}}$, batch size $\beta$ 
 Load the dataset $\mathcal{D}_t^k$, local buffer $\mathcal{M}^k$ 
 Initialize $n = 0$ at the first task 
 **for** each batch $\{(x_i, y_i)\}_{i=1}^{\beta}$ in $\mathcal{D}_t^k$ **do** 
    $g_c = \nabla_w \left[ \frac{1}{\beta} \sum_{i=1}^{\beta} \ell(y_i, f(x_i; w)) \right]$ 
    Sample $\{(x_i', y_i')\}_{i=1}^{\beta}$ from $\mathcal{M}^k$ 
    $g_b = \nabla_w \left[ \frac{1}{\beta} \sum_{i=1}^{\beta} \ell(y_i', f(x_i'; w)) \right]$ 
    $g \leftarrow g_c - \text{proj}_{g_b} g_c \cdot \mathbf{1}(g_b^\top g_c \leq 0)$ 
    $\tilde{g} \leftarrow g - \text{proj}_{g_{\text{ref}}} g \cdot \mathbf{1}(g_{\text{ref}}^\top g \leq 0)$ 
    $w \leftarrow w - \alpha \tilde{g}$ for some learning rate $\alpha$ 
    `ReservoirSampling`$(\mathcal{M}^k, \{(x_i, y_i)\}_{i=1}^{\beta}, n)$ 
    $n \leftarrow n + \beta$ 
 **end for** 
 Return $w$ |

model trained by various baselines, we report a common metric, namely, mean average precision (mAP). This metric assesses the correspondence between the detected bounding boxes and the ground truth, with higher scores indicating better performance. To calculate mAP, we analyze the prediction results obtained from pre-collected driving snippets of vehicular clients. These driving snippets are gathered by navigating the town over a duration of 3000 simulation seconds.

For those experiments on realistic CARLA streaming data, we compare the performances of Ideal, FedAvg, DER and DER+Fed-A-GEM. In the Ideal scenario, the client possesses sufficient memory to retain all data from prior tasks, enabling joint training on all stored data. The last two methods are equipped with buffer size of 200. We train for 70 communication rounds and each round continues for about 200 simulation seconds. The results are presented in Fig. 7b. Note that at communication round 60, one client gets on the highway, which incurs a domain shift. One can confirm that the performance of FedAvg degrades in such domain shift

scenario, whereas DER and DER+Fed-A-GEM maintain the accuracy. Moreover, Fed-A-GEM nearly achieves the performance of the ideal scenario with infinite buffer size, demonstrating the effectiveness of our method.

## E    Continual learning methods with Fed-A-GEM

We provided the pseudocode for Algorithm 2 modifications when implementing FL+DER+Fed-A-GEM and FL+A-GEM+Fed-A-GEM, respectively presented in Algorithm 5 and Algorithm 6. Other FL+CL and CFL methods are also combined with Fed-A-GEM in a similar manner.

Algorithm 5 incorporates Dark Experience Replay (DER) into the local update process on client $k \in [K]$.When the server sends the global model $w$ to client $k$, the client calculates the output logits or pre-softmax response $z$. In addition, the client samples past data $(x', y')$ and the corresponding logits $z'$ from the buffer $\mathcal{M}^k$. To address forgetting, the regularization term considers the Euclidean distance between the sampled output logits and the current model's output logits on buffer data. The gradient $g$ is then refined using this regularization term to minimize the discrepancy between the current and past output logits, thereby mitigating forgetting. The following steps are the same as in the main text.

Algorithm 6 combines with A-GEM, applying gradient projection twice. First, the client computes the gradient $g_c$ with respect to the new data from $\mathcal{D}_t^k$. After replaying previous samples $(x', y')$ stored in the local buffer $\mathcal{M}^k$, the client computes the gradient $g_b$ with respect to this buffered data. If these gradients differ significantly in terms of their direction, the client projects $g_c$ onto $g_b$ to remove interference.

