# OpenReview forum: "Buffer-based Gradient Projection for Continual Federated Learning"
_TMLR — Accepted by TMLR_

### Review · Reviewer_fioA · 2024-10-20

**Summary Of Contributions:**

This submission introduces Fed-A-GEM, a novel adaptation of the A-GEM method tailored for Continual Federated Learning (CFL). By employing a buffer-based gradient projection strategy, Fed-A-GEM effectively mitigates catastrophic forgetting, a prevalent issue where models lose previously acquired knowledge upon learning new information. Unlike existing CFL algorithms, Fed-A-GEM does not rely on predefined task boundaries and leverages local buffer samples along with aggregated buffer gradients to preserve knowledge across diverse clients. This methodology facilitates seamless integration with existing CFL techniques, enhancing their performance substantially. The paper's empirical validations demonstrate considerable performance improvements, with scenarios like task-incremental learning on the CIFAR-100 dataset showing up to a 27% increase in accuracy. The introduction of Fed-A-GEM represents a significant advancement in managing the typical challenges of decentralized, continuous data streams in federated settings, contributing both to academic research and practical applications in federated learning environments.

**Audience:**

Yes

**Claims And Evidence:**

Yes

**Requested Changes:**

1. Inclusion of a Comprehensive Architecture Diagram: Please include a detailed architecture diagram that illustrates the entire workflow and integration points of Fed-A-GEM within the context of Continual Federated Learning (CFL).

2. Analysis of Time Complexity for Secure Aggregation: Amend the section on secure aggregation to include a detailed analysis of its time complexity, along with computational resource requirements. If possible, compare these metrics with other non-secure and secure methods to contextualize the efficiency of Fed-A-GEM.

3. Validation Using More Representative Datasets: Conduct additional experiments using OOD datasets like PACS and Office-Caltech-10.

**Strengths And Weaknesses:**

Strengths:

1. Fed-A-GEM leverages historical data (via shared buffers) from previous tasks, which helps in preserving knowledge without needing to identify task boundaries explicitly.

2. The method is designed to integrate seamlessly with other CFL approaches, enhancing their performance by providing a robust framework against forgetting and improving overall learning stability.

3. Comprehensive experiments have been conducted to verify the effectiveness of Fed-A-GEM across a wide range of classification tasks, including both image and text domains, ensuring the method's versatility.

Weaknesses:

1. Lack of a Comprehensive Architecture Diagram: - The paper does not include an overall architecture figure, which could limit the reader's understanding of the method's workflow and integration points. Such visual aids are crucial for summarizing complex systems and facilitating quicker and clearer comprehension.

2. Inefficiency in Secure Aggregation: - The use of secure aggregation, while beneficial for privacy-preserving purposes, is mentioned to waste time and computational resources. The article fails to provide a detailed analysis of the time complexity associated with this process. An explicit breakdown of the computational costs and the impact on overall system efficiency would be beneficial for evaluating the practical viability of deploying Fed-A-GEM in resource-constrained environments.

3. Utilization of Out of Distribution (OOD) Datasets: - The article employs datasets such as PACS and Office-Caltech-10, which are considered Out of Distribution for many common tasks in federated continual learning. The use of OOD datasets can challenge the generalizability of the findings, as the models might not perform as well when applied to data distributions that are more representative of typical real-world scenarios. Further validation on more standard and diverse datasets could enhance the credibility and applicability of the results.

---

> ### Author Response · Authors · 2024-12-10
> **To Reviewer fioA**
>
> We thank the reviewer for insightful feedback and constructive comments and for providing suggestions to improve our paper.
>
> **Q1. Lack of a Comprehensive Architecture Diagram**
>
> Thanks for your suggestion. We have added this diagram to the paper to help readers to understand the method's workflow.
> https://docs.google.com/document/d/e/2PACX-1vQbf0TBhTNbVW-RX82eCQZQ1l5XtIW5r4B44juVsxRp5s1SB_FCkQ3PwVMcvUjDtDhhIwSM7WHtalEn/pub
>
> **Q2. Analysis of Time Complexity for Secure Aggregation:**
>
> We appreciate the reviewer’s comments regarding the time complexity analysis. To provide a concrete example, we conducted experiments on CIFAR100 over 200 epochs using an NVIDIA A100 GPU. The secure aggregation implementation follows the approach described in [1]. Our results indicate that secure aggregation introduces an additional overhead of 172.01 seconds, representing a 4.23% increase in training time. This overhead consists of two main components: (1) the generation of cryptographic key pairs for all participating clients, and (2) the application of masks to client weights during both model secure aggregation and buffer gradient secure aggregation. We have incorporated these findings into Section 5.1.5 of our manuscript.
>
> [1] Bonawitz K, Ivanov V, Kreuter B, et al. Practical secure aggregation for privacy-preserving machine learning. In proceedings of the 2017 ACM SIGSAC Conference on Computer and Communications Security. 2017.
>
> **Q3. Utilization of Out of Distribution (OOD) Datasets**
>
> We thank the reviewer for the helpful suggestions. To address this concern, we have conducted additional experiments on PACS and Office-Caltech-10.
> * The PACS dataset consists of images across four distinct domains (Photo, Art, Cartoon, Sketch).
> * Office-Caltech-10 contains images of office objects from different sources (Amazon, DSLR, Webcam, Caltech).
>
> For both datasets, we have maintained consistent experimental conditions with our main experiments: ResNet18 (pretrained on ImageNet) as the backbone, 10 clients with non-IID data distribution, and a memory buffer size of 200 per client.
>
> | Methods    | Office-Caltech-10     | PACS      |
> |------------|------------------------|-----------|
> | FL         | 90.47±1.2             | 40.30±2.1 |
> | FL+Ours    | 93.10±0.8 (↑2.63)     | 70.72±1.8 (↑30.42) |
>
> The results demonstrate that Fed-A-GEM significantly improves performance on OOD data, with particularly strong gains on the more challenging PACS dataset (+30.42\%). These results complement our main findings and demonstrate Fed-A-GEM effectiveness in challenging real-world scenarios where distribution shifts are common.
>
> We have incorporated these findings into Section 5.1.10 of our manuscript.

---

### Review · Reviewer_PahG · 2024-11-02

**Summary Of Contributions:**

The paper targets the federated setting where the clients' datasets can change during the course of training. The training dataset is composed of tasks. While previous works focus on the setting where the clients are aware of the task boundaries, the authors in this paper consider a more realistic setting where task boundaries are not explicitly known. Their proposed framework, Fed-A-GEM, is essentially a federated version of a prior A-GEM method. Particularly, in the A-GEM method, gradient over a buffer of samples is used as a reference to project the current gradient, which avoids increasing the model error over the buffer samples. This in turn allows improves the model performance in continual learning. In Fed-A-GEM, the key contribution is that the server obtains the global reference gradient by aggregating the local reference gradients computed by the clients over their local buffers. Each client then utilizes the global reference gradient to remove the undesired component of the local gradients during local model updates. As a result, the global buffer gradient guides the clients towards local model updates that do not hurt the model performance of the buffer samples.

**Audience:**

Yes

**Broader Impact Concerns:**

The reviewer doesn't have any concern that would require a Broader Impact Statement.

**Claims And Evidence:**

Yes

**Requested Changes:**

* While the reviewer acknowledges that Fed-A-GEM provides good performance, it is important for the authors to identify and discuss the algorithmic novelty in more detail.
* Provide experimental results for FL settings where only a random subset of clients participate in each communication round.
* Provide in-depth information for the implementation in Section 5.1.6. For example, it is not clear Class-IL setting, how are the classes incrementally locally and globally?
* The idea of using reference gradients for improving model performance has been previously developed in Byzantine FL works such as [1] and [2]. The authors should provide references to such works.
* In [1], the idea of using weighted aggregation is developed. Specifically, the local update of a client is weighted according to the similarity of the local update with the reference gradient. The authors should provide a weighted variant of Fed-A-GEM to see if that improves performance.

[1] Cao X, Fang M, Liu J, Gong NZ. Fltrust: Byzantine-robust federated learning via trust bootstrapping. arXiv preprint arXiv:2012.13995. 2020 Dec 27.
[2] Prakash S, Hashemi H, Wang Y, Annavaram M, Avestimehr S. Secure and fault tolerant decentralized learning. arXiv preprint arXiv:2010.07541. 2020 Oct 15.

**Strengths And Weaknesses:**

Pros:
* The proposed problem is very interesting.
* The authors explain prior works in detail.
* The paper includes a comprehensive experiment setting.

Cons:
* Novelty of the paper is limited. Particularly, the work is apparently a simple extension of the A-GEM method in a federated setting. The authors do consider the privacy implications, and the proposed Fed-A-GEM is compatible with the standard secure aggregation protocols. However, the paper seems to be a simple combination of relevant prior algorithms.
* Do all clients participate in each round? Assuming that all the clients are always available does not seem very realistic. In most FL settings, only some clients participate in each round.
* This is perhaps fundamental to the general continual learning setting. In a class incremental setting, is the total number of classes known in advance? Is that practical? What if the total classes are not known in advance, how would that change the choice of model architectures?

---

> ### Author Response · Authors · 2024-12-10
> **To Reviewer PahG**
>
> We thank the Reviewer for appreciating the proposed problem, extensive experimental results, and our efforts in making a well-written paper. Our responses are detailed below.
>
> **Novelty of the paper is limited.**
>
> We acknowledge the reviewer's concern that our method may appear simple, as it combines and adapts existing algorithms. While our approach may not be highly unique or novel from a methodological perspective, the extensive experiments, including the new results presented, demonstrate that our method consistently outperforms state-of-the-art techniques.
>
> We strongly believe in the importance of publicizing our work to inform the research community that a simple method can address the Continual Federated Learning problem more efficiently than complex alternatives. This aligned with the **Acceptance Criteria of TMLR** that -- "Crucially, it should not be used as a reason to reject work that isn't considered “significant” or “impactful” because it isn't achieving a new state-of-the-art on some benchmark. Nor should it form the basis for rejecting work on a method considered not “novel enough”, as novelty of the studied method is not a necessary criteria for acceptance. We explicitly avoid these terms (“significant”, “impactful”, “novel”), and focus instead on the notion of “interest”. If the authors make it clear that there is something to be learned by some researchers in their area from their work, then the criterion of interest is considered satisfied. TMLR instead relies on certifications (such as “Featured” and “Outstanding”) to provide annotations on submissions that pertain to (more speculative) assertions on significance or potential for impact."
>
> We kindly request that the Reviewer and Action Editor reconsider the current assessment, taking into account the significant performance improvements our method offers, regardless of its simplicity.
>
> **Do all clients participate in each round? Assuming that all the clients are always available does not seem very realistic. In most FL settings, only some clients participate in each round.**
>
> We appreciate the reviewer’s suggestion to explore more realistic partial client participation settings. We had results on partial client participation in the Appendix C.3, which we have highlighted in blue for your review.
>
> **In a class incremental setting, is the total number of classes known in advance? Is that practical? What if the total classes are not known in advance, how would that change the choice of model architectures?**
>
> Yes, we assume that the total number of classes known in advance in our work. In many research papers and controlled experiments, the total number of classes is often known in advance. However, this is largely unrealistic for real-world applications. In practice, it's often impossible to know the total number of classes that a system will eventually need to handle. Usually they have expandable output layers that can grow as needed.
>
>
> **Provide in-depth information for the implementation in Section 5.1.6. For example, it is not clear Class-IL setting, how are the classes incrementally locally and globally?**
>
> Thanks for your question.
> * The CIFAR-100 dataset is divided into 10 tasks (10 classes each)
> * Data is distributed across 10 clients in a non-IID manner using a Dirichlet distribution ($\alpha=0.3$)
>
> 1. Standard (Synchronous) Setting:
> * All clients finish tasks at the same time
> 2. Asynchronous Setting:
> * Each client processes exactly 500 samples per task
>
> Figure below illustrates the task distribution across users. As shown by the colored bars in the Figure, clients have varying amounts of data per task. The vertical dashed lines indicate 500-sample intervals. In the asynchronous setting, clients may work on different classes and be at different stages of a task compared to others at any given time. We have added details into the Section 5.1.6 and have marked them in blue.
>
> https://docs.google.com/document/d/e/2PACX-1vSA6lxU_UBqOrCjXZtB8448YlMPsOlH2qkckmDyf5hQR2to_RHxabTbsRryzqJjOjt9AtLEZfbP00ol/pub
>
> **The idea of using reference gradients for improving model performance has been previously developed in Byzantine FL works such as [1] and [2]. The authors should provide references to such works.**
>
> Thank you for highlighting these works. In these Byzantine FL works, reference gradients have been explored for securing against malicious updates. Our work leverages reference gradients for preserving knowledge of old tasks in federated learning. We were not aware of this connection between the techniques developed in robust FL and continual federated learning. We have added a description on this connection and have added citations to these Byzantine FL works to acknowledge their use of reference gradients. The relevant additions are marked in blue in the Section 4.

---

> ### Author Response · Authors · 2024-12-10
>
> **In [1], the idea of using weighted aggregation is developed. Specifically, the local update of a client is weighted according to the similarity of the local update with the reference gradient. The authors should provide a weighted variant of Fed-A-GEM to see if that improves performance.**
>
> Thank you for this constructive suggestion. We conducted additional experiments exploring a weighted variant of Fed-A-GEM inspired by FLTrust:
> * Computed cosine similarity scores between each client's local update and global reference gradient
> * Used these scores as weights during model aggregation
>
> Our experiments showed minimal performance differences compared to standard Fed-A-GEM:
>
> #### Effect of weighted aggregation on accuracy (%)
>
> | Dataset          | Fed-A-GEM          | Fed-A-GEM w/ weighted Agg |
> |------------------|--------------------|-----------------------|
> | R-MNIST  (Domain-IL)       | 79.46 ± 4.1        | 77.63 ± 4.8          |
> | P-MNIST  (Domain-IL)      | 34.23 ± 2.7        | 34.28 ± 4.4          |
> | S-CIFAR10  (Class-IL)     | 18.02 ± 0.6        | 18.24 ± 0.4          |
> | S-CIFAR10  (Task-IL)    | 80.83 ± 2.0        | 80.42 ± 3.8          |
> | S-CIFAR100  (Class-IL )    | 17.08 ± 1.8        | 13.46 ± 0.6          |
> | S-CIFAR100  ( Task-IL )    | 74.71 ± 0.9        | 74.74 ± 0.4          |
>
> * Marginal improvements (e.g., 0.22% on sequential-CIFAR10 class-IL)
> * Results suggest the original unweighted approach already effectively preserves knowledge
>
> We have added these findings to the Section 5.1.2.

---

> ### Comment · Reviewer_PahG · 2025-01-04
> **Comment from Reviewer PahG**
>
> Thanks to the authors for sufficiently addressing my queries.

---

### Review · Reviewer_oUGN · 2024-12-04

**Summary Of Contributions:**

This paper proposes a buffer-based approach for federated general continual learning (when task boundaries are not known). The authors have extended an existing Averaged-Gradient Episodic Memory (A-GEM) algorithm to the federated setting.

The clients maintain in their memory, a buffer of relevant past samples, which are used to construct local buffer gradients. The server aggregates them and uses this global buffer gradient to constrain the direction of the update based on the current samples.

The paper is impressive because of the extensive experiments the authors have done.

**Audience:**

Yes

**Claims And Evidence:**

Yes

**Requested Changes:**

Additional explanation for a few things requested above

**Strengths And Weaknesses:**

***Strengths***
- The paper contains extensive experiments, on multiple datasets, studying the impact of multiple factors on the eventual performance. I also appreciate that the authors included limitations of their work in the conclusion.

***Weaknesses***
- This is an empirical work. The authors have extended the A-GEM idea to the federated setting. However, there is not much theoretical justification for many things, beyond some basic intuition. This does not necessarily limit the contribution of the paper. However, I would really appreciate it if the authors could refer to some relevant theory papers (if they exist) on CL.

***Questions***

Given my limited background in CL, I also have a few clarification questions:
- There seem to be two kinds of information - present in model update and past in buffer gradient. Are there CL variants that maintain a more fine-grained distinction in past information, weighting the more recent information more highly?
- The two methods DER (2) and A-GEM (3) solve penalized and constrained versions of the same problem. Under convexity, these are equivalent under mild conditions. But, since convexity is not assumed, how do they compare?
- Section 5.1.1: perhaps domain-IL, class-IL, and task-IL are standard terminology in CL. But, can you explain them in a bit more detail, with an example if possible?
- At the beginning of Section 4, it is said that Fed-A-GEM is ``a federated adaptation of the A-GEM method (Chaudhry et al., 2019).'' However, in Table 1, the 2nd row is FL+A-GEM. This also appears later in the paper. How is this different from Fed-A-GEM?
- Table 1: the difference between w/o Fed-A-GEM and w/ Fed-A-GEM in the rows on FL-L2P is noticeable. Why is the difference so stark for Class-IL, much less for domain-IL, and barely for task-IL? This is also related to the earlier comment about explaining the difference between these three notions.
- Section 5.1.3: in Table 2, the performance of FL-DER on CIFAR10 is a bit surprising, going from $B=200$ to $B=500$. In both cases, accuracy improves for w/o Fed-A-GEM with increasing $B$, but is worse for w/ Fed-A-GEM. Can this be explained somehow?
- Section 5.1.5: I'm not sure of the utility of the 2nd paragraph in this section. Also, the $O(mP)$ complexity does not include the averaging of gradients, while the wall-clock time in seconds does, right?
- Section 5.1.8: in almost all cases in Table 6, w/o Fed-A-GEM accuracy is smaller than in Table 1. But, in multiple cases, w/ Fed-A-GEM accuracy is larger in Table 6 ($K=20$) than in Table 1 ($K=10$). How do we explain this?

---

> ### Author Response · Authors · 2024-12-10
> **To Reviewer oUGN**
>
> We thank the reviewer for their detailed and careful review of our paper and for acknowledging our contribution of extensive experiments. Please find our answers to comments and questions as follows.
>
> **Q0. Refer to some relevant theory papers (if they exist) on CL**
>
> We appreciate the reviewer's suggestion regarding theoretical papers. We have expanded our related work Section 2.1 to include some relevant theoretical papers on Continual Learning, with the work of Bennani et al. [1] being most relevant to our work.
>
> Bennani et al. [1] leverages the Neural Tangent Kernel (NTK) regime, and derives the first generalisation bounds for Stochastic Gradient Descent (SGD) and **Orthogonal Gradient Descent (OGD)** for Continual Learning. This work shows that **OGD** gives a tighter generalization bound than SGD. Kim et al. [2] presents a theoretical framework for understanding and improving Class Incremental Learning (CIL). Their work decomposes CIL into two sub-problems: Within-task Prediction and Task-ID Prediction, and shows that Task-ID Prediction is related to Out-of-Distribution detection. Pentina and Lampert [3] bridge the gap between theory and practice in continual learning by proposing a PAC-Bayesian framework. Their work provides theoretical learning bounds on expected error in future tasks based on the average loss observed in previous tasks. Lee et al. [4] provide a rigorous analysis of the relationship between task similarity and two key phenomena in continual learning: forgetting and transfer.
>
> [1] Bennani M A, Doan T, Sugiyama M. Generalisation guarantees for continual learning with orthogonal gradient descent. arXiv preprint arXiv:2006.11942, 2020.
>
> [2] Kim, G., Xiao, C., Konishi, T., Ke, Z., & Liu, B. A theoretical study on solving continual learning. Advances in Neural Information Processing Systems, 2022.
>
> [3] Pentina A, Lampert C. A PAC-Bayesian bound for lifelong learning. International Conference on Machine Learning. PMLR, 2014.
>
> [4] Lee S, Goldt S, Saxe A. Continual learning in the teacher-student setup: Impact of task similarity. International Conference on Machine Learning. PMLR, 2021.
>
> **Q1. Are there CL variants that maintain a more fine-grained distinction in past information, weighting the more recent information more highly?**
>
> Thank you for this insightful question.  Indeed, the paper [1] introduces biased reservoir sampling, which prioritizes recent data using exponential bias functions. This approach is particularly effective for data streams where recent data holds greater significance. We have incorporated this into our paper Section 5.3.2. We believe these approaches align with and complement our method.
>
> [1] Aggarwal C C. On biased reservoir sampling in the presence of stream evolution. Proceedings of the 32nd international conference on Very large data bases. 2006.
>
> **Q2. The two methods DER (2) and A-GEM (3) solve penalized and constrained versions of the same problem. Under convexity, these are equivalent under mild conditions. But, since convexity is not assumed, how do they compare?**
>
> Thanks for your sharp observation. The reviewer is correct that under convexity, these are equivalent under mild conditions. However, they are not guaranteed to be equivalent in the case of non-convex loss function. Our empirical results show that each method has its strengths in different dataset scenarios. Moreover, it is important to note that our primary contribution is applying Fed-A-GEM to DER and A-GEM, rather than comparing DER and A-GEM.

---

> ### Author Response · Authors · 2024-12-10
>
> **Q3. Section 5.1.1: perhaps domain-IL, class-IL, and task-IL are standard terminology in CL. But, can you explain them in a bit more detail, with an example if possible?**
>
> Thanks for your comments. We explain these three types of incremental learning (IL) settings in Continual Learning (CL) with clear examples and have incorporated this into Appendix A.
>
> 1. Task-IL (Task-Incremental Learning):
> * Each task is clearly labeled and the model knows which task it's performing at test time
> * Example: A model learning to classify:
>     * Task 1: Dogs vs Cats
>     * Task 2: Cars vs Buses
>     * At test time, the model knows "this is a dog/cat classification task" or "this is a vehicle classification task"
> * This is considered simpler because the model can use task-specific parameters/heads since it knows which task it's performing
>
> 2. Domain-IL (Domain-Incremental Learning):
> * Same classes across tasks, but the input distribution changes
> * Example: Classifying digits (0-9) across different domains:
>     * Task 1: Handwritten digits (MNIST)
>     * Task 2: Street view house numbers (SVHN)
> * The classes (0-9) remain the same, but the visual characteristics (domain) change
> * The model needs to learn to recognize the same classes under different conditions/styles
>
> 3. Class-IL (Class-Incremental Learning):
> * New classes are introduced in each task
> * The model must learn to distinguish between all classes seen so far
> * Example: An animal classifier learning:
>     * Task 1: Dog and Cat
>     * Task 2: Bird and Fish
>     * Task 3: Horse and Cow
> * At test time, after learning Task 3, the model must correctly classify any of the 6 animals without knowing which task the input belongs to
> * This is often considered the most challenging setting
>
> **Q4. At the beginning of Section 4, it is said that Fed-A-GEM is ``a federated adaptation of the A-GEM method (Chaudhry et al., 2019).'' However, in Table 1, the 2nd row is FL+A-GEM. This also appears later in the paper. How is this different from Fed-A-GEM?**
>
> We thank the reviewer for the detailed review. Here is a more detailed explanation to clarify their relationships (which has been added to Section 5.1.2):
>
> **FL + A-GEM:** This approach projects the gradient along the **local buffer gradient**, which means that each client independently ensures that its updates do not significantly increase the loss on its locally stored previous data. This helps in mitigating catastrophic forgetting at the client level.
>
> **FL + Fed-A-GEM:** In this method, the gradient is projected along the **aggregated buffer gradient**, which is computed across all clients. This ensures that the updates are in a direction that is favorable for the overall previous data across the federation, providing a more global perspective in mitigating forgetting.
>
> **FL + A-GEM + Fed-A-GEM:** This approach combines the two aforementioned strategies. It first projects the gradient along the local buffer gradient, and then projects it again along the aggregated buffer gradient. This two-step projection aims to balance the need to preserve local knowledge while also aligning with the global knowledge across the federation.
>
> In interpreting their relationship, it is important to consider the trade-offs between local and global knowledge preservation. FL + A-GEM focuses more on preserving local knowledge, FL + Fed-A-GEM emphasizes global knowledge, and FL + A-GEM + Fed-A-GEM seeks to balance the two. The choice between these methods depends on the specific requirements of the federated learning setting and the importance of local vs. global knowledge preservation.
>
> **Q5. Table 1: the difference between w/o Fed-A-GEM and w/ Fed-A-GEM in the rows on FL-L2P is noticeable. Why is the difference so stark for Class-IL, much less for domain-IL, and barely for task-IL? This is also related to the earlier comment about explaining the difference between these three notions.**
>
> We can explain the stark differences:
>
> For Class-IL (Classes are disjoint across tasks):
>
> * The largest gains are seen here because this is the most challenging scenario where:
> 1. The model needs to distinguish between all classes at test time without task identity
> 2. Classes are completely different between tasks, making catastrophic forgetting more severe
>
> For Domain-IL (Same classes, different domains):
>
> * Moderate gains because:
> 1. The fundamental classification task remains the same (same classes)
> 2. Only the domain/distribution changes
> 3. The pre-trained backbone likely has some robustness to domain shifts already
>
> For Task-IL (Known task boundaries):
>
> * Minimal gains because:
> 1. The model knows which task it's performing at test time
> 2. Can use task-specific heads/parameters
>
>
> In short: Fed-A-GEM helps most when the model needs to juggle lots of different classes without hints about which task it's doing (Class-IL), and helps least when the model gets clear information about which task it's handling (Task-IL). We have added this finding into the Section 5.1.2.

---

> ### Author Response · Authors · 2024-12-10
>
> **Q6. Section 5.1.3: in Table 2, the performance of FL-DER on CIFAR10 is a bit surprising, going from $B=200$ to $B=500$. In both cases, accuracy improves for w/o Fed-A-GEM with increasing $B$, but is worse for w/ Fed-A-GEM. Can this be explained somehow?**
>
> Thank you for this detailed observation. Let us analyze the results in detail:
>
> For CIFAR10 (Class-IL), we observe the following patterns:
> * Without Fed-A-GEM: $B=200$ → 18.44% (±3.7) to $B=500$ → 20.81% (±3.6)
> * With Fed-A-GEM: $B=200$ → 30.94% (±3.8) to $B=500$ → 29.78% (±4.3)
>
> The decrease in performance with Fed-A-GEM as $B$ increases is not statistically significant when considering the confidence intervals. Similar patterns are observed in the CIFAR10 (Task-IL) results, where variations also fall within the statistical margin of error.
>
> **Q7. Section 5.1.5: I'm not sure of the utility of the 2nd paragraph in this section. Also, the $O(mP)$ complexity does not include the averaging of gradients, while the wall-clock time in seconds does, right?**
>
> Yes, you're correct. The complexity $O(mP)$ only covers gradient computation at each client, while the wall-clock time measurements include all operations including gradient averaging across clients. We have highlighted this in the Section 5.1.5.
>
> **Q8. Section 5.1.8: in almost all cases in Table 6, w/o Fed-A-GEM accuracy is smaller than in Table 1. But, in multiple cases, w/ Fed-A-GEM accuracy is larger in Table 6 ($K=20$) than in Table 1 ($K=10$). How do we explain this?**
>
> Thanks for your carefull finding!
>
> When there are more users (20 vs 10), usually the performance gets worse because each client has less data to learn from. This is why we see lower accuracy for methods without Fed-A-GEM.
>
> However, Fed-A-GEM actually performs better with 20 users in several cases because:
>
> 1. More users = more diverse stored memories in buffers.
> 2. When combining these diverse memories to guide learning, Fed-A-GEM gets a better overall picture of what knowledge to preserve.
> 3. Each user contributes different perspectives, which helps Fed-A-GEM make better decisions about how to update the model.
>
> So while having more users typically makes things harder, Fed-A-GEM turns this into an advantage by making good use of the diverse experiences from all users. We have added this finding into the Section 5.1.8.
>
>
> We sincerely thank the reviewer for their valuable feedback, which has helped us improve our paper.

---

> > ### Comment · Reviewer_oUGN · 2024-12-21
> > **Comment by the reviewer oUGN**
> >
> > I thank the authors for their detailed response. I have no further questions.

---

### Author Response · Authors · 2024-12-10
**To Action Editor and all Reviewers**

We thank the Reviewers for their insightful feedback and constructive comments for providing suggestions that would improve our paper.

We are encouraged that all reviewers acknowledge the contribution of our extensive, comprehensive, and solid experimental work.

Based on the feedback from the reviewers, we have made the following improvements:
* **Strengthened Empirical Analysis**
    * Conducted new experiments with weighted aggregation variants
    * Included additional evaluations on out-of-distribution datasets (PACS and Office-Caltech-10)
    * Reported time overhead analysis for secure aggregation
* **Enhanced Related Work**
    * Added theory-related work on continual learning
    * Expanded with new literature connected to our ideas
* **Improved Clarity**
    * Added architectural diagrams to better illustrate our approach
    * Clarified experimental settings and results

---

### Decision · Action_Editor_JXis · 2025-01-21

**Recommendation:** Accept as is

**Comment:**

Mostly see above.

Novelty of the algorithmic innovation, while noted, played no part in the individual reviewers' assessments, nor mine.  Even on that front, straightforward combinations of existing methods can still represent novelty (when a new combination or used in a new problem), especially if it gives impressive results on impactful problems.

**Audience:**

The paper represents an important application of ML methods, and the continual learning setting without task boundaries places this work in a more realistic regime.  Very appropriate for the TMLR audience.

**Claims And Evidence:**

The paper does a straightforward combination of existing methods to introduce a new algorithm for Federated Continual Learning, particularly when task boundaries are not known.  The empirical results provides impressive evidence for a dramatic improvement over previous algorithms.  Further results have even been added based on the reviewers suggestions and comments.  The evidence for the claims is strong and comprehensive.